# Temporally resolved early bone morphogenetic protein-driven transcriptional cascade during human amnion specification

Nikola Sekulovski[1], Jenna C Wettstein[1], Amber E Carleton[1], Lauren N Juga[1], Linnea E Taniguchi[1], Xiaolong Ma[2], Sridhar Rao[1,3,4], Jenna K Schmidt[5], Thaddeus G Golos[5,6,7], Chien-Wei Lin[2]*, Kenichiro Taniguchi[1,3]*

[1]Department of Cell Biology, Neurobiology and Anatomy, Medical College of Wisconsin, Milwaukee, United States; [2]Division of Biostatistics, Institute for Health and Equity, Medical College of Wisconsin, Milwaukee, United States; [3]Department of Pediatrics, Medical College of Wisconsin, Milwaukee, United States; [4]Versiti Blood Research Institute, Milwaukee, United States; [5]Wisconsin National Primate Research Center, Milwaukee, United States; [6]Department of Obstetrics and Gynecology, University of Wisconsin - Madison School of Medicine and Public Health, Madison, United States; [7]Department of Comparative Biosciences, University of Wisconsin - Madison School of Veterinary Medicine, Madison, United States

*For correspondence:
chlin@mcw.edu (C-WeiL);
ktaniguchi@mcw.edu (KT)

Competing interest: The authors declare that no competing interests exist.

**Abstract** Amniogenesis, a process critical for continuation of healthy pregnancy, is triggered in a collection of pluripotent epiblast cells as the human embryo implants. Previous studies have established that bone morphogenetic protein (BMP) signaling is a major driver of this lineage specifying process, but the downstream BMP-dependent transcriptional networks that lead to successful amniogenesis remain to be identified. This is, in part, due to the current lack of a robust and reproducible model system that enables mechanistic investigations exclusively into amniogenesis. Here, we developed an improved model of early amnion specification, using a human pluripotent stem cell-based platform in which the activation of BMP signaling is controlled and synchronous. Uniform amniogenesis is seen within 48 hr after BMP activation, and the resulting cells share transcriptomic characteristics with amnion cells of a gastrulating human embryo. Using detailed time-course transcriptomic analyses, we established a previously uncharacterized BMP-dependent amniotic transcriptional cascade, and identified markers that represent five distinct stages of amnion fate specification; the expression of selected markers was validated in early post-implantation macaque embryos. Moreover, a cohort of factors that could potentially control specific stages of amniogenesis was identified, including the transcription factor TFAP2A. Functionally, we determined that, once amniogenesis is triggered by the BMP pathway, TFAP2A controls the progression of amniogenesis. This work presents a temporally resolved transcriptomic resource for several previously uncharacterized amniogenesis states and demonstrates a critical intermediate role for TFAP2A during amnion fate specification.

## eLife assessment

This study presents an **important** dataset that captures the transition from epiblast to amnion using a novel in vitro model of human amnion formation. The supporting evidence for the authors' claims is **convincing**. Key strengths of the study include the efficiency and purity of the cell populations

produced, a high degree of synchrony in the differentiation process, comprehensive benchmarking with single-cell data and immunocytochemistry from primate embryos, and the identification of critical markers for specific differentiation phases. A notable limitation, however, is the model's exclusion of other embryonic tissues.

## Introduction

The amniotic epithelium is the innermost layer of the amniochorionic membrane that physically divides the maternal and fetal environments (*Miki et al., 2005*; *Miki and Strom, 2006*). Amniogenesis is initiated during implantation in humans (and in non-human primates [NHP]), eventually leading to the formation of an amniotic sac structure that surrounds and protects the developing embryo (*Enders et al., 1983*; *Enders et al., 1986*; *Miki et al., 2005*; *Sasaki et al., 2016*; *Shahbazi and Zernicka-Goetz, 2018*; *Taniguchi et al., 2019*). At implantation, the embryo (referred to at this time as a blastocyst) contains three functionally distinct cell types: a collection of unpolarized pluripotent epiblast cells (precursors to the embryo proper and amniotic ectoderm), a surrounding layer of polarized trophectoderm (a placental tissue precursor), and an underlying extraembryonic primitive endoderm (a yolk sac precursor). During implantation, the pluripotent epiblast cells undergo apico-basal polarization to form a cyst with a central lumen, the future amniotic cavity (*Carleton et al., 2022*). This event is followed by the fate transition of pluripotent epiblast cells at the uterine-proximal pole of the cyst to squamous amniotic ectoderm, forming a sharp boundary between amnion and pluripotent epiblast portions of the cyst (*Shahbazi and Zernicka-Goetz, 2018*; *Shao and Fu, 2022*; *Taniguchi et al., 2019*). This structure, the amniotic sac, represents the substrate for the next essential steps of embryonic development. Recent transcriptomic studies of early human and NHP embryos have presented benchmark resources for the amnion transcriptome of primates (*Bergmann et al., 2022*; *Nakamura et al., 2016*; *Nakamura et al., 2017*; *Sasaki et al., 2016*; *Tyser et al., 2021*; *Yang et al., 2021*).

The mechanistic dissection of human amnion development presents a challenge, since it is ethically unacceptable to manipulate human embryos in vivo and technically difficult to carry out such studies in vitro. NHP embryos such as rhesus macaque are a suitable alternative to human embryos, due to their close ancestral similarities (*Nakamura et al., 2021*), and have already provided crucial insights into molecular processes that could potentially regulate amniogenesis. Indeed, genetic manipulation has been demonstrated in NHP systems to study peri-implantation embryogenesis (*Yang et al., 2021*), although this is not yet a widely applicable approach due to resource availability and logistical complexity of such studies, which also carry their own unique ethical challenges.

Previously, we developed an in vitro system of human amnion development called Gel-3D (*Shao et al., 2017a*; *Shao et al., 2017b*), in which human pluripotent stem cells (hPSC) are plated on a soft gel substrate consisting of extracellular matrix (ECM). Within 24 hr, aggregates of hPSC form polarized cysts composed of columnar pluripotent cells. Over the next 3–4 days, a 3D environment is introduced by supplementing the culture medium with a diluted ECM overlay. Under these conditions, the initially pluripotent cysts spontaneously undergo squamous morphogenesis, lose pluripotency markers, and begin to express amnion markers; in over 90% of such cysts, the central lumen becomes entirely surrounded by squamous amnion cells (hPSC-amnion, *Shao et al., 2017a*). Using this system, we showed that mechanically activated bone morphogenetic protein (BMP) signaling can trigger amniogenesis and identified several genes that are now used as amnion markers (e.g. GATA3, TFAP2A, GABRP, *Shao et al., 2017a*; *Shao et al., 2017b*). These findings were supported by later studies using early human embryo-like models as well as by single-cell transcriptomic studies in early human and NHP embryos (*Chen et al., 2021*; *Tyser et al., 2021*; *Yang et al., 2021*; *Zheng et al., 2019*; *Zheng et al., 2022*).

Though this Gel-3D system (*Shao et al., 2017a*) has been valuable in allowing initial investigations of the mechanisms underlying amniogenic differentiation, it suffers from the fact that the initiation of amniogenesis by mechanical cues is stochastic and is not fully penetrant, hindering reproducible examinations into temporally and spatially sensitive processes. To better implement mechanistic studies, we developed a highly reproducible hPSC-based amniogenic system, called Glass-3D$^{+BMP}$, that does not rely on the mechanically activated spontaneous BMP signaling to initiate amniogenesis. Rather, in Glass-3D$^{+BMP}$, BMP4 ligand is added exogenously, and the amniotic differentiation response is quickly

seen in all cells. Interestingly, while BMP signaling activation is immediately detectable in all cells, squamous morphogenesis (starting approximately 24 hr after BMP4 addition) is initially focal within a given cyst, but then spreads laterally, to form fully squamous amnion cysts by 48 hr. Such amnion cysts (hPSC-amnion) are uniformly composed of mature amniotic cells that share similar transcriptomes with amnion cells from a Carnegie stage (CS)7 human embryo (dataset from *Tyser et al., 2021*).

Given the robust and synchronous nature of this system, we perform detailed time-course bulk RNA sequencing analyses of developing Glass-3D$^{+BMP}$ hPSC-amnion to comprehensively characterize changes in transcriptional characteristics during the initial stages of amnion fate specification. Strikingly, from 0 to 48 hr after BMP treatment, developing hPSC-amnion displays transcriptomic characteristics of epiblast, intermediate, or amnion cells that are present in the CS7 human embryo, revealing a potential amnion differentiation trajectory. Moreover, we systematically characterize dynamically expressed genes, and identify immediate-, early-, intermediate-, and late-responding genes associated with amniotic differentiation, some of which are validated and characterized in detail in NHP embryos. Interestingly, in contrast to the previous idea that TFAP2A is a *pan*-amnion marker expressed throughout amniogenesis, we establish that TFAP2A is an early/intermediate-responding gene, with a temporal pattern of activation that is distinct from immediate-responding genes such as GATA3, another *pan*-amnion marker. Functional analyses show that, in the absence of TFAP2A, amniogenesis is incomplete, suggesting that TFAP2A controls amnion fate progression. Lastly, we identify a population of cells at the boundary of the amnion and the epiblast in the peri-gastrula macaque embryo that shares striking molecular and cellular characteristics of cells undergoing amnion specification in Glass-3D$^{+BMP}$. Together, using the Glass-3D$^{+BMP}$ amnion platform, these studies present a transcriptomic map of the early stages of amnion differentiation, and identify a critical transcriptional pathway dependent on TFAP2A, as well as a potential site of active amniogenesis in the primate peri-gastrula.

## Results

### Generation of fully squamous hPSC-derived amniotic cysts in a reproducible and controlled manner using a Glass-3D$^{+BMP}$ culture system

Targeted mechanistic investigations into amnion fate specification has been limited by the fact that amniogenesis is not fully penetrant in existing in vitro embryo-like models (*Chen et al., 2021*; *Nasr Esfahani et al., 2019*; *Shao et al., 2017a*; *Shao et al., 2017b*; *Zheng et al., 2019*), prompting us to explore alternative strategies. Since our previous studies have established that BMP signaling is a major driver of amniogenesis (*Shao et al., 2017a*; *Shao et al., 2017b*), we sought to establish a system in which we could directly control the activation of BMP signaling. We began with a culture condition called Glass-3D, in which aggregates of hPSC are cultured in a stiff glass substrate condition with the culture medium supplemented with a 3D ECM overlay (*Figure 1A*). Under these conditions, aggregates of hPSC initiate radial organization and form cysts with a central lumen within 48 hr (*Shao et al., 2017a*; *Taniguchi et al., 2017*; *Taniguchi et al., 2015*). Importantly, these cysts are composed entirely of polarized pluripotent cells. When we examined the polarized membrane spatial proteomics of these hPSC-pluripotent cysts, we found that BMP receptors are enriched at the basolateral membrane territory (*Wang et al., 2021*). To test whether BMP signaling could be directly activated in these cysts by the addition of BMP to the medium, pluripotent cysts (derived from H9 [WA09] human embryonic stem cells for all experiments, unless otherwise noted) were treated with exogenously provided BMP4 at d2 (*Figure 1B*, *Figure 1—video 1*, Glass-3D$^{+BMP}$). Nuclear localization of phosphorylated SMAD1/5 (pSMAD1/5), as detected by immunofluorescence (IF), was used as a readout for BMP signaling activation (*Figure 1C*). At d2 in Glass-3D, prior to the addition of BMP, a polarized pluripotent (SOX2$^+$) cyst structure has formed, and a small central lumen is visible (*Figure 1C* – 0 min); this structure is similar to that of the developing epiblast immediately prior to the initiation of amniogenesis (*Shahbazi and Zernicka-Goetz, 2018*; *Taniguchi et al., 2019*). Strikingly, within 10 min after BMP addition, uniform nuclear pSMAD1/5 enrichment is seen in all treated cells at an equivalent level (*Figure 1C* – 10 min, fluorescent quantitation in *Figure 1D*). This is quickly followed by the expression of GATA3, a *pan*-amnion marker and a well-known immediate downstream BMP target that has been shown to facilitate pluripotency exit (*Gunne-Braden et al., 2020*). GATA3 expression is detected in all cells within 3 hr (*Figure 1E*), suggesting that BMP signaling is uniformly activated. Interestingly, TFAP2A, another

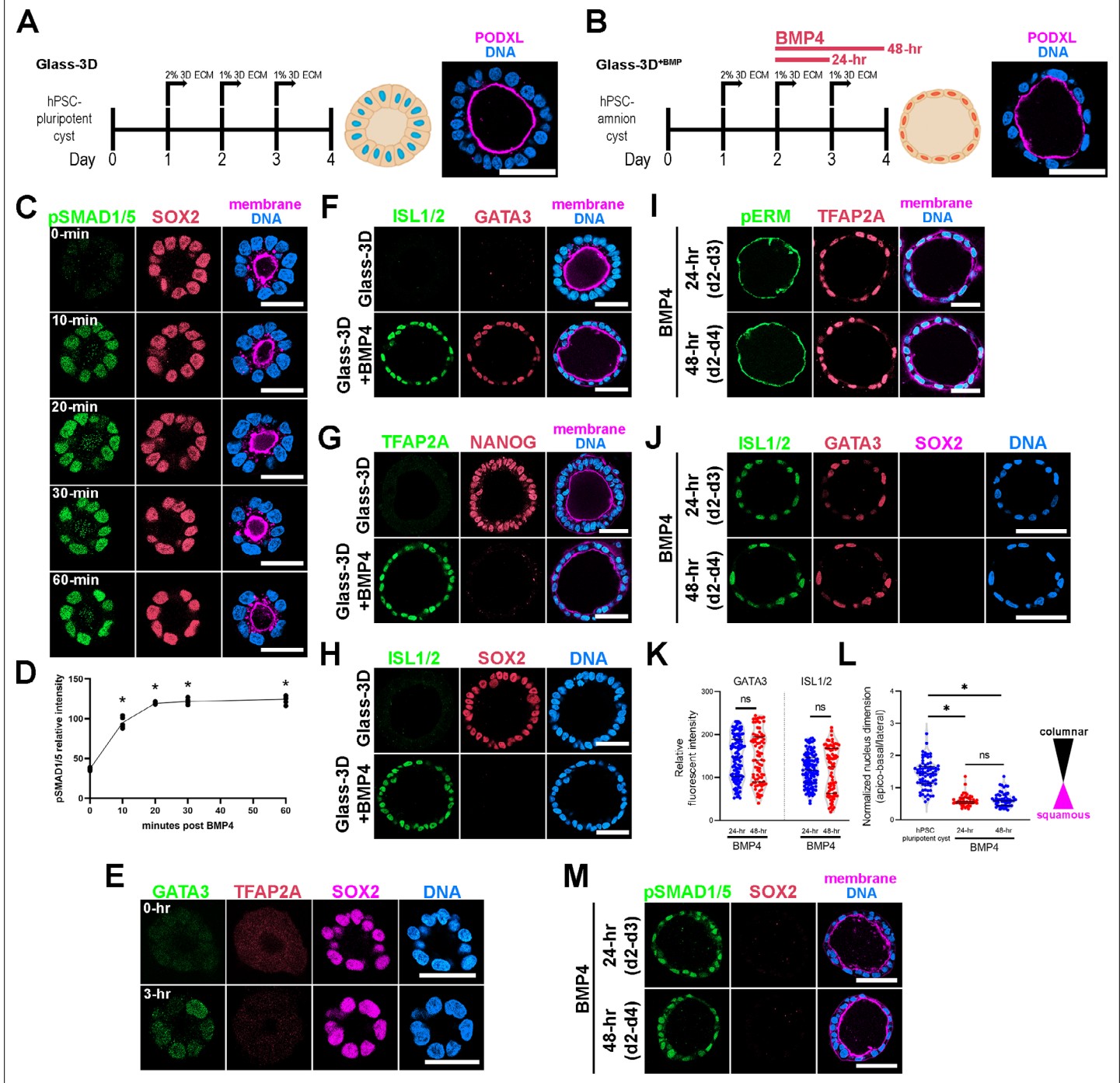

**Figure 1.** Highly penetrant amniogenesis in a Glass-3D⁺BMP culture system. (**A,B**) Human pluripotent stem cells (hPSC)-derived pluripotent cyst (hPSC-pluripotent cyst, Glass-3D in **A**) and amnion cyst (Glass-3D⁺BMP in **B**) assays: immunofluorescence (IF) images show hPSC-pluripotent (**A**) and -amnion (**B**) cysts stained for PODXL (lumen, magenta) and DNA (nuclear shape, blue). BMP4 treatment (20 ng/mL) leads to uniformly squamous amnion cysts by day 4. (**C**) Confocal optical sections of d2 cysts harvested prior to (0 min) and 10, 20, 30, and 60 min after BMP4 treatment, stained with indicated markers. Staining for phosphorylated SMAD1/5 reveals a prominent nuclear pSMAD1/5 enrichment within 10 min of BMP4 treatment in all cells. (**D**) Quantification of nuclear pSMAD1/5: five independent samples were counted, more than 50 cells were counted per sample. * indicates statistically significant changes ($p < 0.05$) compared to 0 min timepoint. (**E**) Abundant GATA3 expression (in green), but not TFAP2A (red), is seen by 3 hr after BMP4 treatment while SOX2 expression is maintained. (**F–H**) Pluripotency (NANOG, SOX2) and amnion (ISL1/2, GATA3, TFAP2A) marker expression analyses of d4 cysts grown in the Glass-3D (top) or in Glass-3D⁺BMP (bottom) conditions. (**I, J**) Confocal micrographs of d4 cysts grown in the Glass-3D⁺BMP system stained for phosphorylated EZRIN, RADIXIN, and MOESIN (pERM), an apical membrane marker, as well as with indicated amnion markers (top – cysts were treated with BMP4 for 24 hr between d2 and d3; bottom – treated with BMP4 for 48 hr from d2 to d4). (**K, L**) Quantification of GATA3 and ISL1/2

*Figure 1 continued on next page*

Figure 1 continued

fluorescent intensities in 24 (blue, n=115 cells) and 48 hr (red, n=76 cells) BMP4 treatment samples (**K**), as well as of nuclear aspect ratios of d4 hPSC-cyst (black, n=63 cells), and 24 (blue, n=45 cells) and 48 hr (red, n=52) BMP4-treated cysts (harvested at d4), (**L**). (**M**) pSMAD1/5 staining (green) shows similar nuclear enrichment between 24 and 48 hr BMP4 treatment at d4 (also stained with indicated markers). Scale bars = 50 μm. BMP, bone morphogenetic protein.

The online version of this article includes the following video for figure 1:

**Figure 1—video 1.** All optical sections of Glass-3D+BMP human pluripotent stem cell (hPSC)-amnion shown in *Figure 1B*.

https://elifesciences.org/articles/89367/figures#fig1video1

transcription factor thought to be an early *pan*-amnion marker, is not detectable in GATA3 positive cells 3 hr after BMP addition (*Figure 1E*). After 48 hr of BMP4 treatment, all cysts show uniform squamous morphogenesis, loss of pluripotency markers (NANOG, SOX2), and activated expression of additional *pan*-amnion markers, TFAP2A and ISL1 (*Figure 1F–H*).

To investigate whether exogenous BMP4 ligand is continuously needed throughout the 48 hr time-course in order to form fully squamous hPSC-amnion, d2 pluripotent cysts were cultured with BMP4 for 24 hr (d2–d3), then cultured without BMP4 for the following 24 hr (d3-d4), and then harvested at d4 (4 days after initial plating, *Figure 1B*). Interestingly, these 24 hr BMP treatment samples show fully squamous amnion cysts that are morphologically and molecularly similar to those treated over 48 hr. That is, they show uniformly squamous cyst morphogenesis and prominent expression of GATA3, TFAP2A, and ISL1 (*Figure 1I and J*, quantitation for fluorescent intensity and nuclear aspect ratios in *Figure 1K and L*), as well as relatively similar nuclear pSMAD1/5 signals (*Figure 1M*). Moreover, in the 24 hr BMP treatment samples (*Figure 2A*), time-course analyses show that heterogeneous GATA3 expression is visible in a few nuclei at 3 hr and 6 hr, becoming more prominent by 12–24 hr (*Figure 2B–D*). Early GATA3 expression is followed by initiation of TFAP2A expression by 12 hr (*Figure 2B and D*). Two other '*pan*-amnion' genes, *ISL1* and *HAND1,* become activated by 24 hr (*Figure 2C and E*), revealing a previously unrecognized GATA3 → TFAP2A → ISL1/HAND1 amniotic gene activation order (*Figure 2B and C*, mRNA expression levels in *Figure 2D and E*). Importantly, SOX2 expression is abundant at 6 hr, fades in some nuclei at 12 hr, and is largely diminished by 24 hr (*Figure 2B and C*, mRNA expression levels in *Figure 2F*). NANOG expression follows a similar time-course to that of SOX2, revealing a gradual loss of pluripotency (*Figure 2F*). Samples treated with BMP4 for 24 hr or 48 hr display similar levels of amnion and pluripotency markers (*Figure 2G*), suggesting that BMP4 treatment may be dispensable after the first 24 hr. To test this possibility, we treated cysts with BMP4 for 24 hr (d2-d3) and subsequently treated with LDN-193189, a small molecule inhibitor for the BMP receptor kinase activity (d3-d4). This treatment resulted in fully squamous cysts that are positive for GATA3 and ISL1/2 (*Figure 2H*), confirming that BMP signaling initiates the early amniogenesis cascade, but is not required for its progression.

Together, these results show that BMP4 treatment of d2 pluripotent cysts in a Glass-3D culture environment provides a robust amniogenic model that enables mechanistic investigations. We confirmed this finding in a human induced pluripotent stem cell line (1196a, *Figure 2—figure supplement 1A and B*, *Chen et al., 2014*) as well as in H7 human ESC (*Figure 2—figure supplement 1C*), further establishing this protocol as an effective model of amnion fate conversion in vitro. Given that BMP4 treatment from d3 to d4 is dispensable, 24 hr BMP4 treatment between d2 and d3 is used in all remaining experiments using Glass-3D+BMP, unless otherwise noted.

## Transcriptomic characterization of developing Glass-3D+BMP hPSC-amnion

To comprehensively understand the transcriptomic changes during hPSC-amnion development in the Glass-3D+BMP system, we performed bulk RNA sequencing analyses for samples taken at 0, 0.5, 1, 3, 6, 12, 24, and 48 hr timepoints (post-BMP4 treatment). Samples were also taken from pluripotent monolayers and d4 pluripotent cysts (*Figure 3A* [experimental outline], *Figure 3B* [heatmap], *Figure 3C* [principal component analysis], see *Supplementary file 1A* for normalized and filtered counts). Overall, a total of 12,891 genes were detected among all sequenced samples. Similar to known immediate BMP-responding genes (e.g. *ID2*, *BAMBI*), *GATA3* shows a significant increase by 1 hr, and continuously increases over time (*Figure 3D*, >100-fold increase by 1 hr). Other markers also

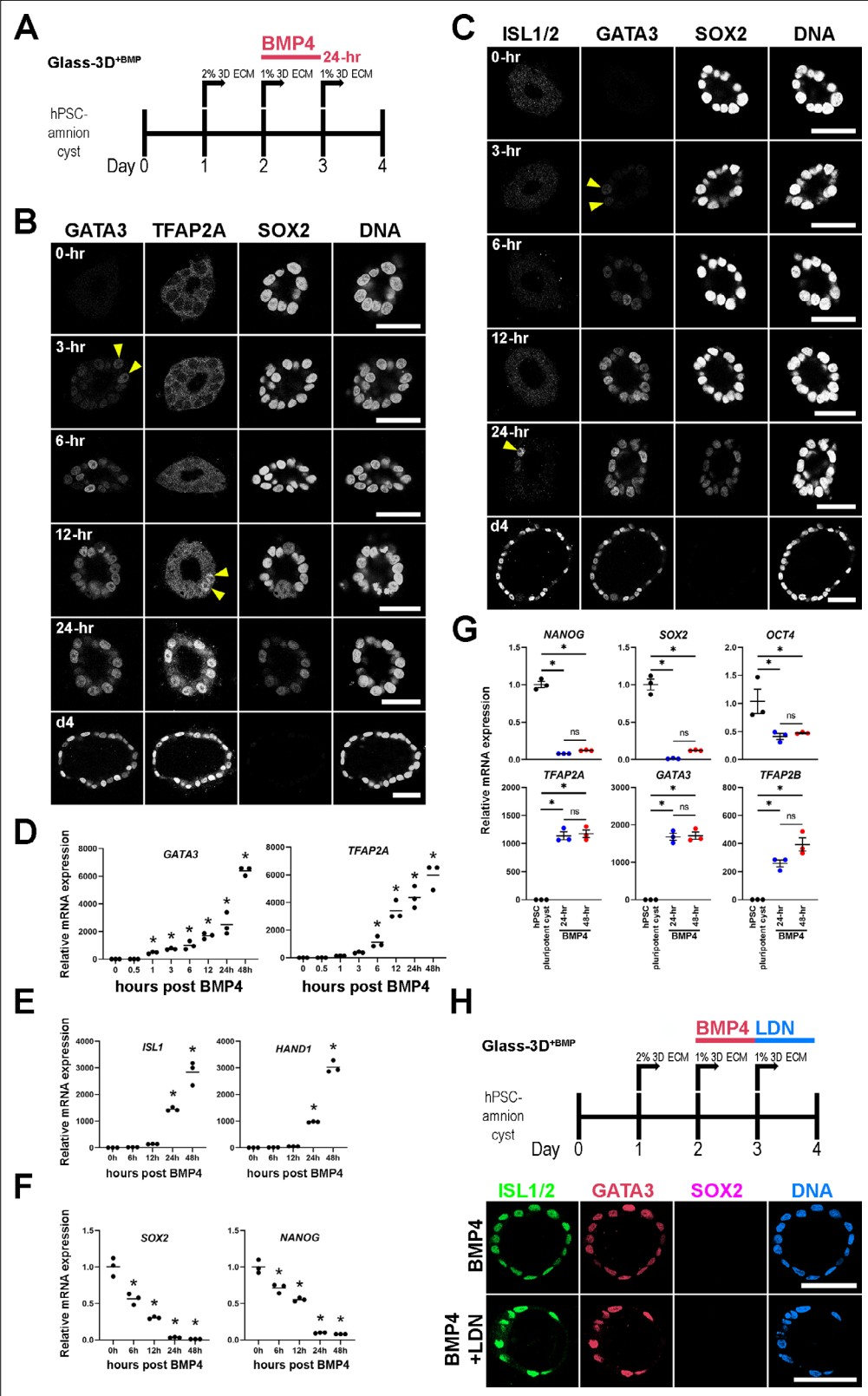

**Figure 2.** Time-course analysis of Glass-3D[+BMP] human pluripotent stem cell (hPSC)-amnion development. (**A**) Timeline of the Glass-3D[+BMP] hPSC-amnion assay. Twenty four hr BMP treatment is sufficient to generate uniformly squamous hPSC-amnion cysts. (**B,C**) Optical sections of time-course Glass-3D[+BMP] hPSC-amnion cysts (harvested at indicated timepoints after BMP4 treatment) stained with indicated markers (inverted black and white

*Figure 2 continued on next page*

*Figure 2 continued*

fluorescent signals shown for individual channels, to aid visualization). GATA3, an immediate BMP target is seen by 3 hr (yellow arrowheads). hPSC-amnion cysts with prominent nuclear TFAP2A (**B**) and ISL1/2 (**C**) are seen by 12 and 24 hr, respectively (yellow arrowheads), while SOX2 shows gradual decrease. Consistent changes are also observed at mRNA transcript levels using quantitative RT-PCR (**D–F**), also see the expression of additional markers: *HAND1* (amniotic) and *NANOG* (pluripotent); * indicates statistically significant changes (p<0.05) compared to 0 hr timepoint. (**G**) Relative mRNA expression analyses of indicated pluripotency (top) and amniotic (bottom) markers between d4 Glass-3D hPSC-pluripotent cyst (black), and d4 Glass-3D$^{+BMP}$ hPSC-amnion with 24 hr (blue) or 48 hr (red) BMP treatment; * indicates statistically significant changes (p<0.05). (**H**) BMP inhibitor (LDN-193189) treatment outline as well as confocal micrographs of d4 Glass-3D$^{+BMP}$ hPSC-amnion (top, control – 24 hr BMP4 treatment between d2 and d3) and hPSC-amnion treated with LDN-193189 between d3 and d4 after BMP4 treatment (between d2 and d3). Scale bars = 50 µm. BMP, bone morphogenetic protein.

The online version of this article includes the following figure supplement(s) for figure 2:

**Figure supplement 1.** Glass-3D$^{+BMP}$ human pluripotent stem cell (hPSC)-amnion formation using 1196a human induced pluripotent stem cell and H7 human embryonic stem cell lines.

---

display expected expression patterns (*Figure 3E–G*). For example, *TFAP2A* and *ISL1* show significant increases by 3 hr and 24 hr, respectively (*Figure 3E*), and pluripotency markers (*Figure 3G*, *NANOG*, *OCT4/POU5F1*, *SOX2*) show reduction over time. A comparison of gene expression between d4 Glass-3D pluripotent cysts and d4 Glass-3D$^{+BMP}$ hPSC-amnion (48 hr timepoint) shows 1820 upregulated and 2092 downregulated (cutoff=2-fold, p<0.05) genes in d4 Glass-3D$^{+BMP}$ hPSC-amnion (*Figure 3H*, see *Supplementary file 1B and C* for full lists). Highly upregulated markers include known advanced amnion state markers such as *GABRP*, *IGFBP3*, and *EPAS1* (*Figure 3H*), and highly downregulated genes include *SOX2* and *NANOG* (*Figure 3H*, see *Supplementary file 1D and E* for Gene Ontology [GO] enrichment analyses [Biological Processes] for all significantly upregulated and downregulated genes [>2-fold, FDR<0.05]).

The transcriptomic characteristics of developing amnion cells from a CS7 human embryo were recently described by Tyser et al. (*Figure 4A* shows a UMAP [Uniform Manifold Approximation and Projection] plot with the original coordinates and annotations, *Tyser et al., 2021*). In this analysis, the authors state that yolk sac and connecting stalk tissues were discarded during dissection, but amnion tissue was intact and included in the single-cell preparation that was analyzed (*Tyser et al., 2021*). Consistent with this notion, our analysis in which expression was superimposed on the UMAP plot shows that *pan*-amnion markers such as *GATA3*, *TFAP2A*, and *ISL1* are broadly expressed in the rod-shaped cell population marked 'Ectoderm' (*Figure 4B*, areas within the inset 'i' are shown in *Figure 4B–D*; see *Figure 4—figure supplement 1A* for uncropped UMAP expression plots), suggesting that this region contains cells of the amniotic lineage. We next examined which cells in the CS7 human embryo share transcriptomic similarities to developing hPSC-amnion, and found that, indeed, the 48 hr Glass-3D$^{+BMP}$ samples exhibit clear transcriptional similarities to cells marked 'Ectoderm' in the CS7 human embryo (*Figure 4C*), validating the human amnion-like characteristics of our hPSC-amnion model. Interestingly, the earlier timepoints also show overlaps with cell types of the CS7 human embryo (*Figure 4C*, uncropped plot shown in *Figure 4—figure supplement 1B*). While pluripotent monolayer, d4 Glass-3D pluripotent cyst, as well as 0, 0.5, 1, 3, and 6 hr samples transcriptomically resemble Tyser 'Epiblast' cells (orange dots in *Figure 4C*), 12 hr samples show transcriptomic similarities with cells labeled in the Tyser analysis as 'Primitive Streak'; these cells map close to the Epiblast/Primitive Streak boundary in the Tyser UMAP (green dots in *Figure 4C*). By 24 hr, the transcriptomes are most similar to Tyser 'Primitive Streak' cells proximal to the amniotic 'Ectoderm' (blue dots in *Figure 4C*). Despite their 'Primitive Streak' annotation, these 24 hr cells adjacent to the amniotic 'Ectoderm' population display relatively low levels of *TBXT* (a major marker of primitive streak cells, *Figure 4D*, *Beddington and Robertson, 1999*; *Tyser et al., 2021*; *Zheng et al., 2019*), marking them as distinct from the *TBXT*$^{high}$ 'Primitive Streak' cells progressing to 'Nascent Mesoderm' (*Figure 4D*). Indeed, in developing hPSC-amnion, while not detectable using IF (*Figure 4—figure supplement 1C*), a *TBXT*$^{low}$ transcriptional state is seen starting at the 24 hr timepoint (*Figure 4—figure supplement 1D*). A similar trajectory as well as gene expression patterns are also seen in the *Yang et al., 2021*, cynomolgus macaque peri-gastrula single-cell RNA sequencing dataset (*Figure 4E–G*, uncropped plots shown in *Figure 4—figure supplement 1E and F*). These bioinformatic analyses suggest that

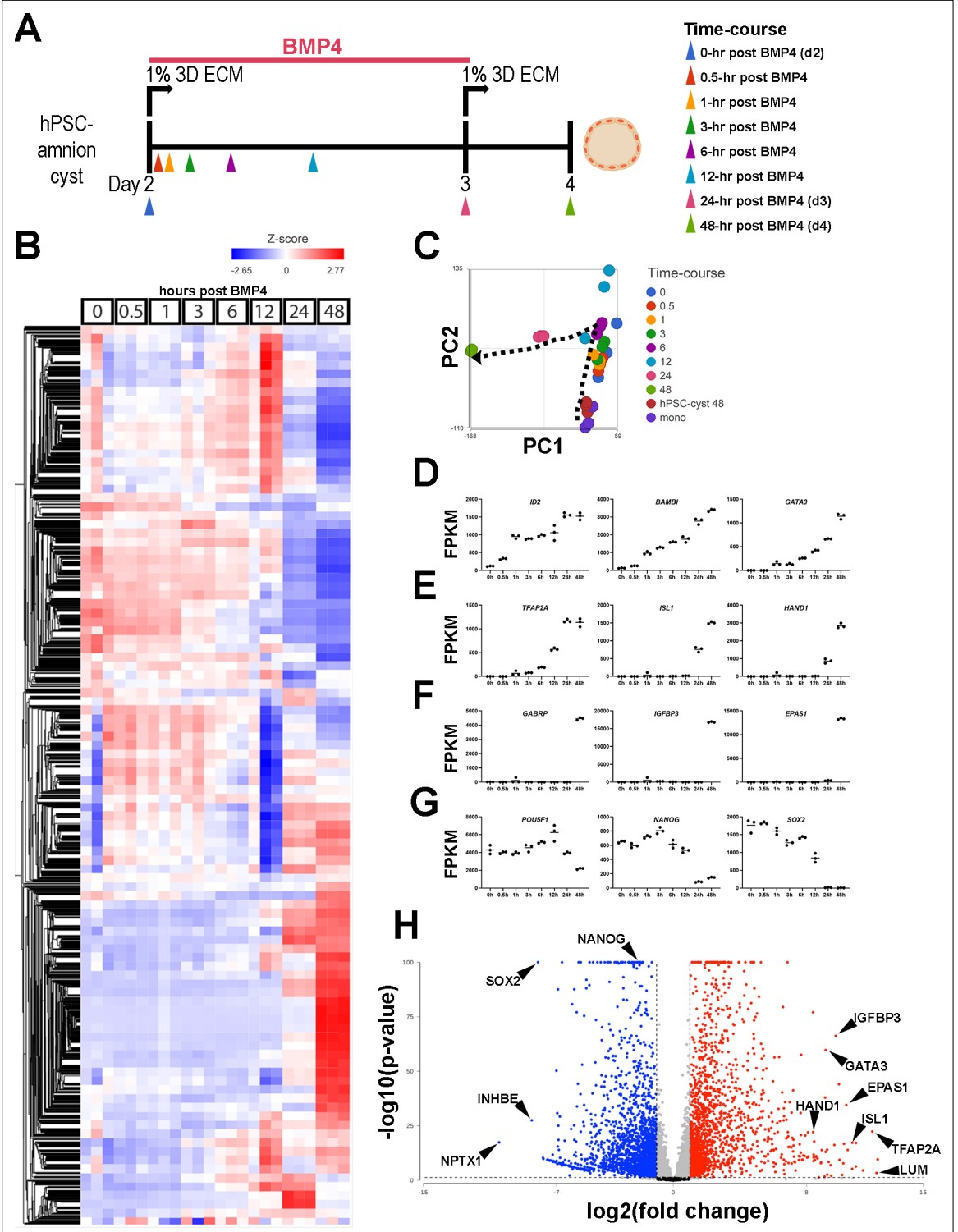

**Figure 3.** Temporally resolved transcriptomic characterization of Glass-3D⁺ᴮᴹᴾ human pluripotent stem cell (hPSC)-amnion. (**A**) Experimental timeline (from d2) of the time-course bulk RNA sequencing analysis. Colored arrowheads indicate timepoints at which samples were harvested. (**B**) A heatmap showing a hierarchical clustering analysis of all detected genes (12,940): bulk transcriptomes from triplicate samples are shown at each timepoint. A gradient scale indicates Z-score. (**C**) Principal component analysis (PCA) plot of time-course bulk transcriptomes, using the most variable genes, revealing progressive transcriptomic changes over time (indicated by the dotted arrow). Note that coloring is consistent with the arrowheads in (**A**). (**D–G**) Time-course plots of normalized expression values for selected known immediate BMP target (**D**), general amnion (**E**), advanced amnion (**F**), and pluripotency genes (**G**). (**H**) A volcano plot analysis showing up- (red) and down- (blue) regulated genes in Glass-3D⁺ᴮᴹᴾ hPSC-amnion cysts (gene

*Figure 3 continued on next page*

*Figure 3 continued*
expression compared to d4 pluripotent hPSC-cysts): black lines represent cutoffs for p-value<0.05 and twofold difference. BMP, bone morphogenetic protein.

the Glass-3D⁺ᴮᴹᴾ amnion may develop from pluripotent epiblast-like cells that are transcriptomically equivalent to the CS7 epiblast cells, and, then traverse *TBXT*ˡᵒʷ intermediate states as the lineage specification continues before acquiring an amniotic fate by 48 hr (dotted arrow, *Figure 4C*).

Given that the hPSC-amnion differentiation trajectory in the Glass-3D⁺ᴮᴹᴾ model overlaps with cells in the CS7 human embryo, it appears that active amniogenesis processes may be present in the early post-implantation embryo. Thus, we applied an RNA velocity analysis (*Bergen et al., 2020*) to the Tyser data; in such an analysis, lineage progression is inferred from comparing spliced and unspliced mRNA. *Figure 5A* (inset 'i') shows that the Tyser velocity data show differentiation trajectories very similar to the one inferred from the time-course analysis of the Glass-3D⁺ᴮᴹᴾ model.

To further validate this notion in an NHP system, we performed detailed IF expression analyses for GATA3, TFAP2A, ISL1, and SOX2 in an early cynomolgus macaque embryo staged between CS6 and CS7 (*Figure 5B–E*). In this CS6/7 embryo, a clear amniotic sac is seen, with SOX2⁺ epiblast cells on one side and contiguous ISL1⁺ squamous amniotic ectoderm cells that are also positive for GATA3 and TFAP2A on the other (*Figure 5B–E*, fluorescent intensity quantitation shown in *Figure 5C, E*). Surface ectoderm bridging neural ectoderm and amniotic ectoderm is not seen at this stage; this finding is consistent with the previous histological studies in human (the Carnegie Collection, found in the Virtual Human Embryo Project) and NHP pre- and peri-gastrula embryos (*Enders et al., 1983*; *Enders et al., 1986*; *Yang et al., 2021*). Although amniotic mesenchyme is not evident, some mesenchymal cells are observed underlying the epiblast (*Figure 5F*, green dotted bracket). These mesenchymal cells show abundant nuclear pSMAD1/5 (*Figure 5F*) and TBXT (*Figure 4—figure supplement 1G*); these cells are likely derived from the primitive streak. Strikingly, at the boundary between the epiblast and amnion, a gradual transition in cell morphology is readily seen: while cells most proximal to the epiblast are columnar, there is a gradual reduction in height until fully squamous cells are seen (*Figure 5B, D, F, and H*), represented by the changes in nuclear aspect ratio (*Figure 5H*). Molecularly, the expression of GATA3, TFAP2A, ISL1, and SOX2 is correlated with such columnar to squamous cell shape change (*Figure 5B–E*). While SOX2 expression shows a reducing trend, the expression of GATA3, TFAP2A, and ISL1 gradually increases in more distal squamous cells (*Figure 5B–E*, quantitation in *Figure 5C, E*). Furthermore, GATA3, TFAP2A, and ISL1 each display distinct expression patterns within the boundary cell population, as the expression of GATA3 and TFAP2A is first detected in columnar SOX2⁺ cells, but in these cells, ISL1 is not observed (*Figure 5C*). However, a weak but clear expression of ISL1 can be found within ~10 distally located cells, from the first detected GATA3⁺/TFAP2A⁺ cell. Similar observations are seen in another early cynomolgus macaque embryo (*Figure 5—figure supplement 1*, staged at CS7). This cell shape change associated with the ordered transcriptional events (from GATA3, TFAP2A then ISL1) is similar to the ordered morphological and molecular changes seen in developing Glass-3D⁺ᴮᴹᴾ hPSC-amnion. Thus, the results based on these bioinformatic and expression analyses provide evidence for the presence of amniotic fate progression in early post-gastrula embryos at the boundary between amnion and epiblast (a schematic model shown in *Figure 5I*).

## Gene expression dynamics in developing Glass-3D⁺ᴮᴹᴾ hPSC-amnion

To identify key players potentially involved in amnion fate specification at different stages, we comprehensively characterized all significantly upregulated genes based on when significant upregulations are seen: 'immediate' (*Supplementary file 2A*, up in 0.5 hr or 1 hr samples), 'early' (*Supplementary file 2B*, 3 or 6 hr), 'intermediate-1' (*Supplementary file 2C*, 12 hr), 'intermediate-2' (*Supplementary file 2D*, 24 hr), and 'late' (*Supplementary file 2*E, 48 hr); genes that were kept after a more stringent filtering (described in Materials and methods) are shown in *Figure 6*, enabling the identification of additional temporal markers.

Previous studies have identified several *pan*-amnion markers that are known immediate BMP targets (e.g. *GATA3* and others in the list of 'immediate' genes, *Supplementary file 2A*), which are expressed in other early BMP-driven tissue morphogenesis. Among the known *pan*-amnion markers, ISL1 is a highly promising marker for early lineage specified amnion cells (*Yang et al., 2021*; *Zhao et al., 2021*; *Zheng et al., 2022*). In accord with this, we found that ISL1 expression is limited to squamous amnion

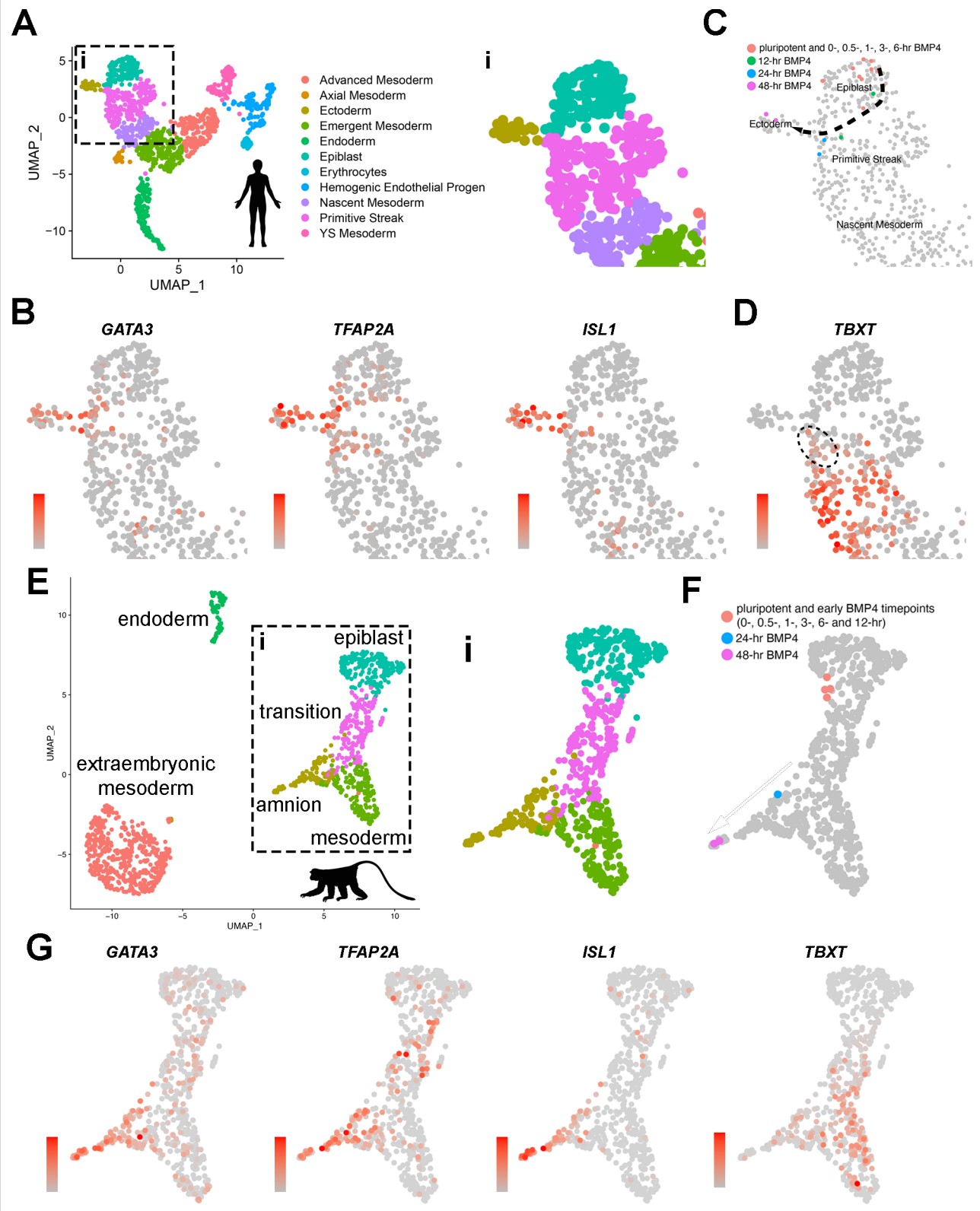

**Figure 4.** Transcriptome comparison of developing Glass-3D⁺ᴮᴹᴾ human pluripotent stem cell (hPSC)-amnion cysts, a Carnegie stage (CS)7 human embryo, and two GD14 cynomolgus macaque peri-gastrula. (**A**) A Uniform Manifold Approximation and Projection (UMAP) plot displaying the original single-cell transcriptome coordinates of the CS7 human embryo described in Tyser et al., shown with the original annotations. The inset (**i**) is used for expression analyses in (**B**). (**B**) Expression of amnion markers – *GATA3, TFAP2A,* and *ISL1* – superimposed onto the UMAP plot in (**A – i**).

*Figure 4 continued on next page*

*Figure 4 continued*

(**C**) Transcriptome similarity analysis comparing the CS7 human embryo single-cell dataset and the Glass-3D^+BMP time-course bulk RNA sequencing dataset, showing the human CS7 embryo UMAP plot (inset) with colored cells that are transcriptomically similar to each timepoints as indicated (orange – pluripotent cysts/monolayer and 0, 0.5, 1, 3, and 6 hr post-BMP4; green – 12 hr post-BMP4; blue – 24 hr; magenta – 48 hr). While pluripotent and earlier timepoints (orange colored cells) overlap with Epiblast cells, some 12 hr (green) and all 24 hr (blue) samples are similar to a subpopulation of Primitive Streak cells, and 48 hr (magenta) samples overlap exclusively with Ectoderm (see Materials and methods for detailed bioinformatics pipeline). (**D**) *TBXT* expression transposed onto the CS7 human embryo UMAP plot, revealing *TBXT*^low subpopulation that show transcriptomic similarities to 24 hr post-BMP samples (indicated by dotted circle). (**E**) An integrated UMAP plot for two in vitro cultured gestation date (GD) 14 (8 days after culturing 6-day-old blastocyst) cynomolgus macaque peri-gastrula (replotted and reclustered from Yang et al.). (**F**) Transcriptome similarity analysis comparing the GD14 cynomolgus macaque peri-gastrula dataset, and the Glass-3D^+BMP time-course dataset, showing the Yang et al. GD14 UMAP plot (inset 'i') with colored cells that are transcriptomically similar to each timepoints as indicated (orange – pluripotent cysts/monolayer and 0, 0.5, 1, 3, 6, and 12 hr post-BMP4; blue – 24 hr; magenta – 48 hr). While pluripotent and early datapoints (orange colored cells) overlap with epiblast cells, the 24 hr samples share similarities with *TBXT*^low population, and the 48 hr samples overlap with ISL1^high amnion cells (**G**), expression analysis of indicated markers. Gradient scales indicate expression level (orange = high, gray = low). Uncropped plots are found in *Figure 4—figure supplement 1*. BMP, bone morphogenetic protein. Created with BioRender.com.

The online version of this article includes the following figure supplement(s) for figure 4:

**Figure supplement 1.** Original Uniform Manifold Approximation and Projection (UMAP) plots and TBXT expression analyses.

cells and is not seen in the epiblast or in transitioning boundary cells (*Figure 5B*). Notably, in Glass-3D^+BMP, *ISL1* first shows a significant upregulation at 24 hr (intermediate-2, *Figure 7A*). This correlates with the timeframe at which squamous cells are first observed (*Figure 2*). This finding suggests that amnion specification is likely initiated prior to 24 hr in Glass-3D^+BMP and that genes that are upregulated between 12 hr and 24 hr could be used as additional markers for amnion specification. Consistent with the studies by Tyser et al., we identified *DLX5* (716.76-fold increase between 12 hr and 24 hr, and 2.4-fold increase from 24 hr to 48 hr, *Figure 7B*). Interestingly, however, our expression analysis of the Tyser et al. dataset reveals a broader *DLX5* expression pattern compared to *ISL1* (the *DLX5* expression domain map is similar to that of *TFAP2A*, *Figure 4B*). Indeed, our IF analyses for DLX5 in developing hPSC-amnion as well as in a CS7 cynomolgus macaque embryo show expression patterns similar to TFAP2A (*Figure 7C and D*). These results further validate ISL1 as a marker for specified amnion cells and establish that DLX5 as an additional marker for cells undergoing amniogenesis as well as mature amnion cells.

## Functional tests of an immediate-, GATA3, and an early-, TFAP2A, responding transcription factors

Immediate- and early-responding transcription factors are functionally relevant candidates that initiate amnion specification and drive the expression of early amnion lineage specified markers such as ISL1. Among the genes in the immediate- and early-responding categories, we screened for genes associated with transcription, and identified several genes, including immediate – GATA3 and early – TFAP2A. As noted above, GATA3 expression precedes TFAP2A expression by approximately 9 hr. Both of these factors show highly enriched expression, and both have previously established functions as pioneer transcription factors (*Rothstein and Simoes-Costa, 2020*; *Takaku et al., 2016*; *Tanaka et al., 2020*; *Van Otterloo et al., 2022*; *Zaret and Carroll, 2011*). Therefore, we examined the role of GATA3 and TFAP2A by generating hPSC-amnion in Glass-3D^+BMP derived from H9 cells lacking GATA3 or TFAP2A (two clonal lines with unique mutations were used for each analysis, see *Figure 8—figure supplement 1* for KO validation). Interestingly, *GATA3*-KO hPSC grown in Glass-3D^+BMP develop into fully squamous cysts that express amnion markers, perhaps suggesting functional redundancy. In contrast, while controls show uniformly squamous amniotic cells at 48 hr, *TFAP2A*-KO cysts contain both squamous (ISL1^+) and columnar (NANOG^+/SOX2^+) cell populations (*Figure 8A* [Glass-3D hPSC-pluripotent cysts], *Figure 8B and C* [Glass-3D^+BMP hPSC-amnion cysts]). Indeed, a doxycycline (DOX)-inducible expression of TFAP2A in the KO background leads to control-like squamous morphogenesis as well as uniform expression of GATA3 and ISL1 by 48 hr (d4, *Figure 8C*). These results suggest that amniogenesis may be halted in the absence of *TFAP2A*. To examine this further at transcriptomic levels, we performed time-course bulk RNA sequencing for control and *TFAP2A*-KO hPSC-amnion

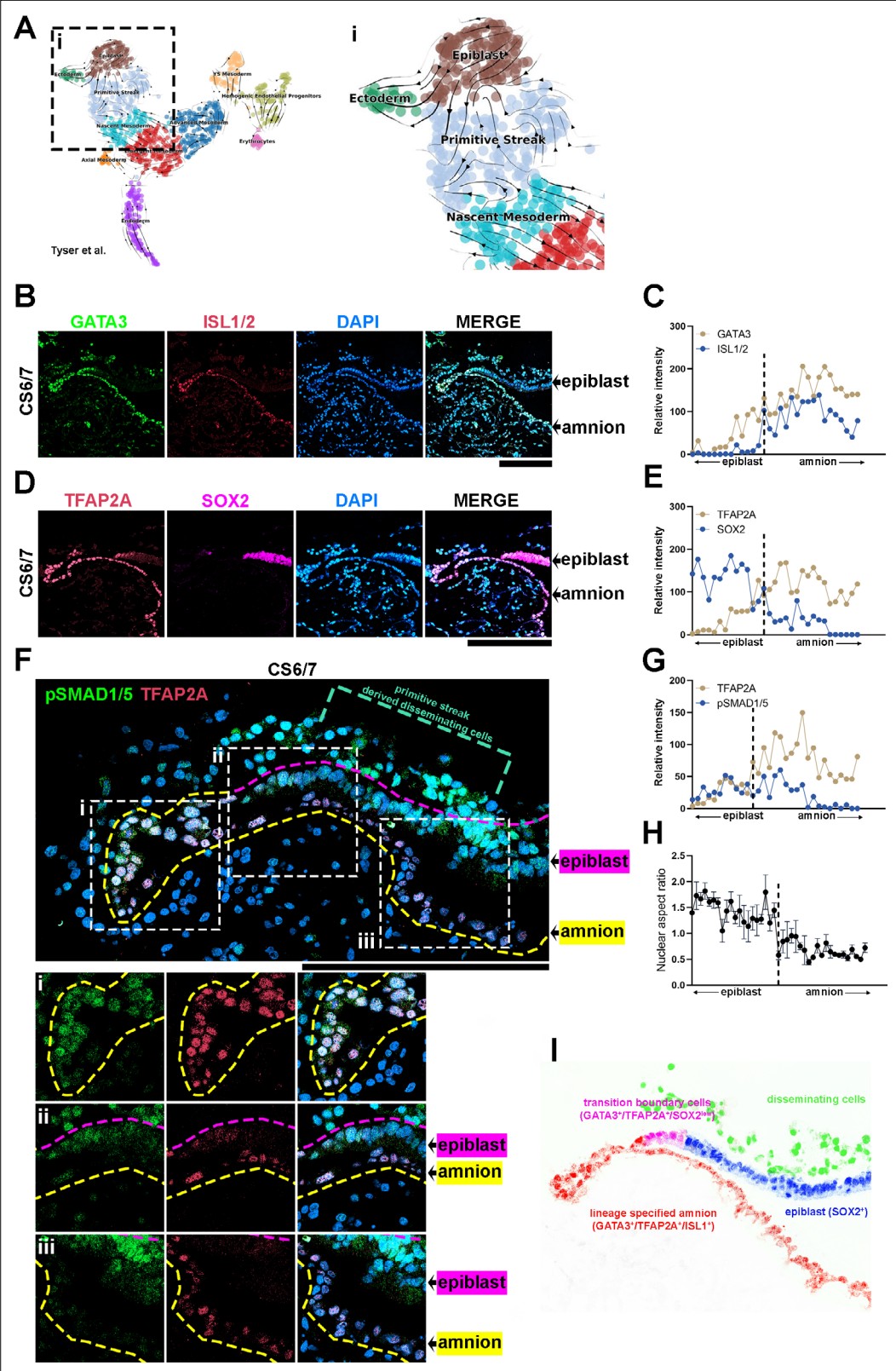

**Figure 5.** Transcriptional characterizations of amnion-epiblast squamocolumnar boundary of the Carnegie stage (CS)6/7 cynomolgus monkey amniotic sac. (**A**) RNA velocity analysis (scVelo) of the Tyser et al. CS7 human embryo dataset, showing predicted lineage progression trajectories from Epiblast to Ectoderm traversing the *TBXT*^low population. (**B–G**) Optical sections of a CS6/7 cynomolgus macaque embryo stained for indicated transcription

*Figure 5 continued on next page*

*Figure 5 continued*

factors (GATA3, ISL1/2, pSMAD1/5, and SOX2, **B, D, F**), as well as quantitation for relative fluorescent intensity in proximal (closer to epiblast) to distal (closer to amnion) cells of indicated markers at the amnion-epiblast boundary (**C**, **E**, and **G** are quantitation of images in **B**, **D**, and **F**, respectively). Each dot represents a single cell. (**i**), (**ii**), and (**iii**) indicate insets in (**F**). Half of the amniotic sac shown. Quantitation shows that, while GATA3⁺/TFAP2A⁺ cells are also SOX2⁺, ISL1⁺ cells are only seen after SOX2 expression is no longer detected: ISL1 expression is exclusive to squamous amnion cells. Moreover, the expression of pSMAD1/5 and TFAP2A is reduced in most distal amnion (**ii**, **iii**). Scale bars = 200 μm. (**H**) Nuclear aspect ratio quantitation of amnion-epiblast boundary cells in (**B**), (**C**), and (**D**), mean + standard error of the mean (SEM) shown. Dotted vertical line in each plot indicates amnion-epiblast boundary (**C, E, G, H**). (**I**) A schematic representation of the CS6/7 cynomolgus macaque amniotic sac: the confocal micrographs in (**B**) were edited to represent SOX2⁺ epiblast (pseudocolored in blue), primitive streak-derived disseminating (green), GATA3⁺/TFAP2A⁺/SOX2^low transition boundary (magenta) and GATA3⁺/TFAP2A⁺/ISL1⁺ lineage specified amnion (red) cells.

The online version of this article includes the following figure supplement(s) for figure 5:

**Figure supplement 1.** Quantitation for relative fluorescent intensity in proximal and distal cells of indicated markers at the amnion-epiblast boundary in a Carnegie stage (CS)7 cynomolgus macaque embryo.

at 0, 12, 24, and 48 hr, and performed a transcriptome similarity analysis using the *Tyser et al., 2021* human CS7 embryo dataset. Indeed, while an expected Epiblast to Ectoderm differentiation trajectory similar to *Figure 4C* is found in controls (*Figure 8D*, left), *TFAP2A*-KO samples do not overlap with Ectoderm cells, even at 48 hr (*Figure 8D*, right). Rather, 48 hr *TFAP2A*-KO samples show transcriptomic characteristics similar to some Primitive Streak cells (*Figure 8D*, uncropped plots shown in *Figure 8—figure supplement 2*), and, strikingly, our IF analysis shows that some cells robustly expressing TBXT⁺ are seen in *TFAP2A*-KO cysts at 48 hr (*Figure 8E*). These results reveal a paused

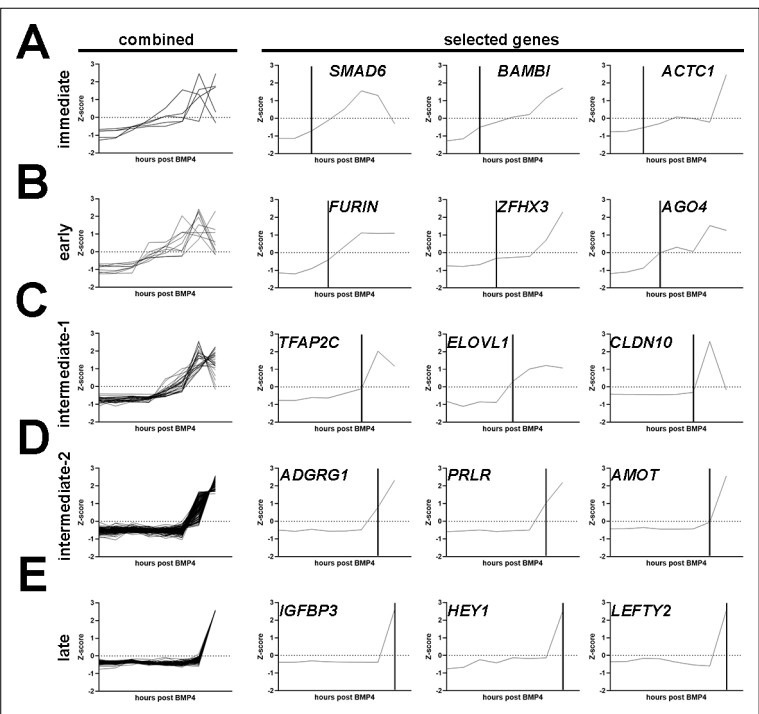

**Figure 6.** Characterization of differentially expressed genes identified through the Glass-3D⁺ᴮᴹᴾ human pluripotent stem cell (hPSC)-amnion time-course sequencing analysis. (**A–E**) Combined and selected genes are shown for each category (**A** – immediate; **B** – early; **C** – intermediate-1; **D** – intermediate-2; **E** – late). Genes for each category are selected based on significant upregulation (FDR<0.05, fold >2, based on Z-scores) by 1 (immediate, n=5), 3 (early, n=9), 6–12 (intermediate-1, n=30), 24 (intermediate-2, n=178), and 48 hr (late, n=157) post-BMP4 treatment, compared to the previous timepoint. Vertical lines indicate timepoints at which significant changes are seen. BMP, bone morphogenetic protein.

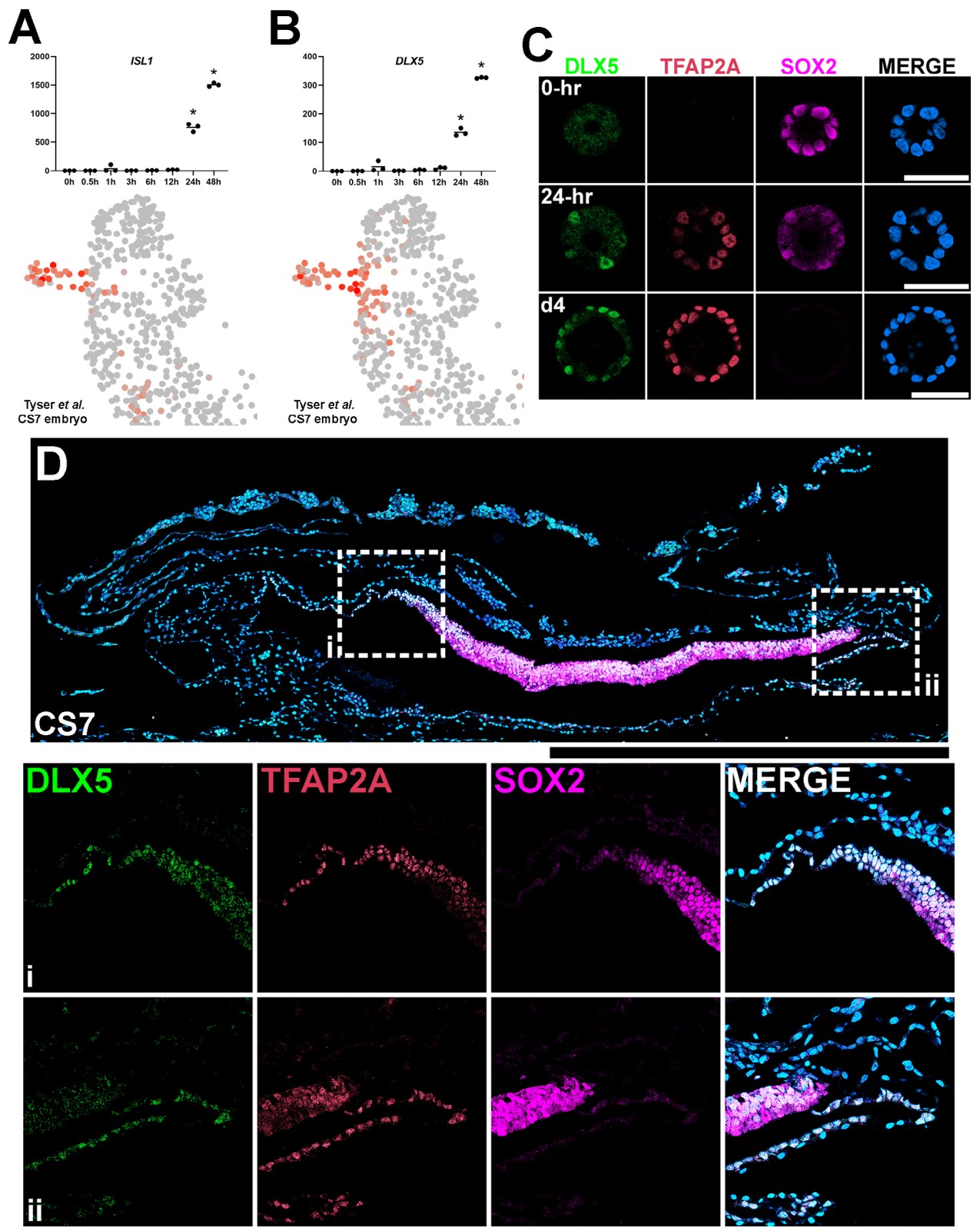

**Figure 7.** Establishment of DLX5 as a new marker of amniogenesis similar to TFAP2A. (**A,B**) Expression analyses for *ISL1* (**A**) and *DLX5* (**B**) in the time-course bulk RNA sequencing (top) and in the single-cell Carnegie stage (CS)7 human embryo (bottom) datasets. (**C**) Confocal optical sections of Glass-3D^+BMP human pluripotent stem cell (hPSC)-amnion cysts (at 0 hr, 24 hr (**d3**) and d4) stained with indicated markers. (**D**) Histological section of a cynomolgus monkey amniotic sac at CS7 stained with indicated markers (SOX2^+ epiblast; TFAP2A^+/DLX5^+/SOX2^- amnion). Insets (**i and ii**) indicate

*Figure 7 continued on next page*

*Figure 7 continued*

the amnion-epiblast boundaries. DLX5 and TFAP2A share a similar expression pattern at the boundary. Similar to pSMAD1/5 and TFAP2A (shown in *Figure 5F*), most distal amnion cells show reduced DLX5 expression. Surface ectoderm bridging neural ectoderm and amniotic ectoderm is not seen at this stage. Amniotic mesenchyme is observed underlying the TFAP2A+/DLX5+ amniotic epithelium at this stage. Scale bars = 500 μm.

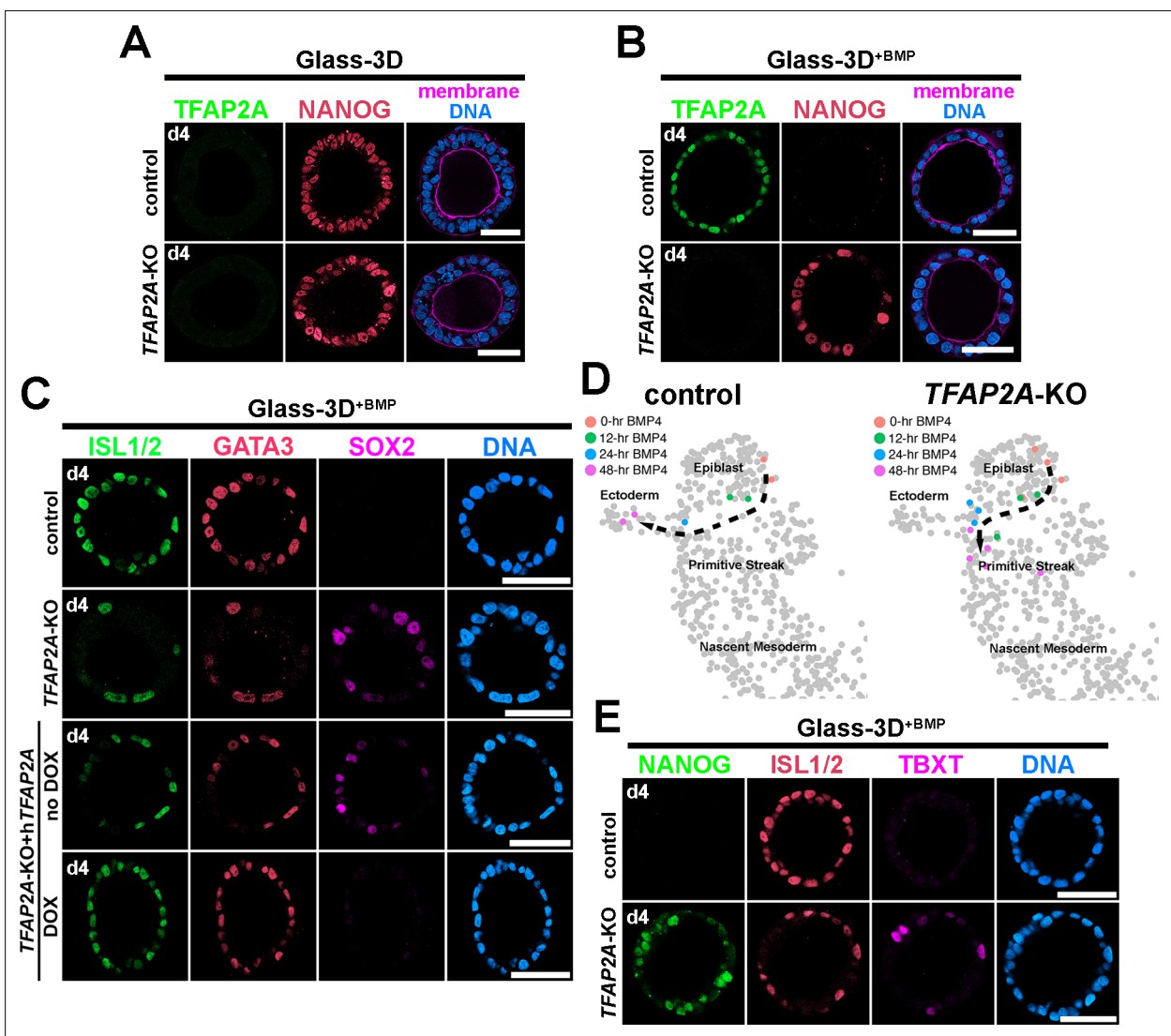

**Figure 8.** Analyses of Glass-3D+BMP human pluripotent stem cell (hPSC)-amnion in control and *TFAP2A*-KO background. (**A,B**) Optical sections of control and *TFAP2A*-KO cysts in Glass-3D (**A**, pluripotent) as well as in Glass-3D+BMP (**B**). While control-like pluripotent cysts are formed in Glass-3D, several NANOG+ cells are still seen in Glass-3D+BMP in the absence of *TFAP2A*. (**C**) Confocal micrographs of d4 control, *TFAP2A*-KO and *TFAP2A*-KO carrying a doxycycline (DOX)-inducible human TFAP2A transgenic construct with or without DOX treatment, stained with indicated markers. (**D**) Transcriptome similarity analysis comparing the Carnegie stage (CS)7 human embryo single-cell dataset with the time-course (0, 12, 24, and 48 hr post-BMP) bulk dataset from control and *TFAP2A*-KO Glass-3D+BMP hPSC-amnion. Orange, green, blue, and magenta colored cells are transcriptomically similar to 0, 12, 24, and 48 hr post-BMP bulk RNA sequencing samples, respectively. While a clear differentiation trajectory from Epiblast to Ectoderm is seen in control (left), a halted amnion lineage progression is seen in the *TFAP2A*-KO background. (**E**) Optical sections of control and *TFAP2A*-KO cysts in Glass-3D+BMP stained for NANOG (green), ISL1/2 (red), and TBXT (magenta). Scale bars = 50 μm. BMP, bone morphogenetic protein.

The online version of this article includes the following figure supplement(s) for figure 8:

**Figure supplement 1.** Validation of *GATA3*- and *TFAP2A*-KO human pluripotent stem cell (hPSC) lines.

**Figure supplement 2.** Uncropped transcriptional similarity Uniform Manifold Approximation and Projection (UMAP) plots for control and *TFAP2A*-KO shown in *Figure 8D*.

amniogenesis phenotype in the absence of TFAP2A, providing a previously unrecognized notion that TFAP2A is critical for maintaining amnion fate progression.

## Discussion

In this study, we developed a reproducible and faithful model for robust mechanistic examinations of human amnion fate specification, and identified a BMP4-dependent amniogenic transcriptional cascade, as well as a role for TFAP2A in maintaining amnion fate progression. Moreover, we established DLX5 as an additional marker for lineage progressing amnion cells and provided evidence for an active amniogenic transcriptional cascade at the epiblast/amnion boundary in peri-gastrula human and NHP embryos. These findings and the temporally resolved amniogenic gene lists provided in this study offer additional avenues for further exploration of human amniogenesis.

Previous studies established several broad amnion markers (e.g. GATA3, TFAP2A, and ISL1) that are expressed in developing, as well as, differentiated amnion cells. However, comprehensive studies have not been performed to fully characterize how each of these markers corresponds to amnion fate specification steps, especially since BMP signaling is a primary trigger of amniogenesis and a large set of BMP target genes are activated. In this study, a time-course transcriptomic analysis was employed to identify genes that display temporal-specific dynamics among genes that show more static expression patterns: this enabled us to define 'stage-specific' genes that can be used to distinguish amniogenic fate progression states. In particular, we found that *GATA3* is an immediate BMP target during amniogenesis (along with other known targets such as *ID1*, *ID2,* and *BAMBI*); a similar BMP4 → GATA3 transcriptional pathway has been well established previously in other systems (*Abe et al., 2021*; *Bonilla-Claudio et al., 2012*; *Gunne-Braden et al., 2020*; *Vincentz et al., 2016*).

Early activation of GATA3 is followed by the expression of second-tier 'early' genes such as *TFAP2A* by 6 hr (clear TFAP2A protein is detected by 12 hr). Our functional analyses indicate that loss of TFAP2A results in incomplete amniogenesis, with failure to downregulate pluripotent genes, loss of some differentiated amnion markers, and, in some cells, increased expression of TBXT, a primitive streak marker. Thus, TFAP2A, a previously recognized pioneer factor (*Rothstein and Simoes-Costa, 2020*; *Van Otterloo et al., 2022*), appears to be an important component that ensures the proper progression of the amniogenesis process, perhaps by suppressing the TBXT$^{high}$ primitive streak-like state.

Interestingly, a recent study by Castillo-Venzor et al. reported that, in contrast to our findings, amniogenesis appears intact in the absence of *TFAP2A* (*Castillo-Venzor et al., 2023*). However, there are multiple differences between the system utilized by Castillo-Venzor et al. vs. the one reported here. Castillo-Venzor et al. employ an hPSC-based primordial germ cell-like cell (PGCLC) differentiation culture protocol, in which amniotic and mesodermal cells are also formed. In that system, embryoid bodies are generated from hPSC treated with Activin A and CHIR (a small molecule WNT activator) to form 'pre-mesodermal cells', and then are grown in the presence of exogenous BMP, SCF (stem cell factor, encoded by *KITLG*), EGF (epidermal growth factor), and LIF (leukemia inhibitory factor) as floating aggregates. Moreover, while uniform amniogenesis is seen in Glass-3D$^{+BMP}$ (a characteristic that enables robust mechanistic examinations specifically into amnion fate progression), the PGCLC differentiation culture used by Castillo-Venzor et al. leads to the formation of cells of multiple lineages. Therefore, the loss of TFAP2A in that system may be compensated by signaling factors provided by mesodermal and/or PGCLC. Alternatively or additionally, it is possible that, in this PGCLC culture, while initial amniogenesis might be present (similar to the focal amniogenesis seen in our system), the later fate spreading event may be absent, perhaps due to differences in tissue geometry (aggregates versus cysts), or due to differences in degrees of lineage commitment by the time that spreading can occur. Together, these results suggest that a single lineage culture system (lacking additional cell types), such as the one we describe here, may more readily expose certain critical events and, therefore, in some cases, may more robustly enable mechanistic investigations into lineage progression.

Several hours following the activation of *TFAP2A*, third-tier 'intermediate' genes such as *ISL1* (ISL1 protein by 24 hr: only in squamous cells) are expressed. It has been previously established that *ISL1* is a key amnion marker in early primate embryos (*Yang et al., 2021*; *Zhao et al., 2021*; *Zheng et al., 2022*). Interestingly, our IF analyses of early peri-gastrulation cynomolgus macaque embryos show that ISL1 expression is restricted to amnion cells that are already squamous in nature (*Figure 5*). Additionally, our time-course transcriptomic analyses in Glass-3D$^{+BMP}$ reveal that *ISL1* expression is seen at

12–24 hr, later than *TFAP2A* activation but much earlier than activation of more differentiated amnion markers such as *GABRP*, *EPAS1*, and *IGFBP3* (48 hr). Together, these results suggest that amnion cells progress from GATA3+, to GATA3+/TFAP2A+, to GATA3+/TFAP2A+/ISL1+, to mature amnion cells that express these three markers as well as the more mature amnion markers such as GABRP, EPAS1, and IGFBP3.

The first amniogenesis event is seen upon implantation, in the epiblast cells underlying the polar trophectoderm, establishing the amniotic sac, an asymmetric amnion-epiblast cystic structure (*Enders et al., 1983*; *Enders et al., 1986*; *Shahbazi and Zernicka-Goetz, 2018*; *Taniguchi et al., 2017*). In this study, we provide evidence for the active progression of amniogenic fate conversion at the amnion-epiblast boundary of the cynomolgus macaque peri-gastrula. What is the difference between amniogenesis that occurs during implantation and during gastrulation? A previous study by *Sasaki et al., 2016*, showed that, in a macaque embryo staged at PreG1 (just initiating implantation), a stage before *BMP4* expression is seen in the amnion at PreG2, *BMP2* is abundantly expressed in the visceral endoderm. However, the visceral endoderm underlies the epiblast population that gives rise to the embryo proper (not amnion), and *CER1* is also abundantly expressed in the visceral endoderm (*Sasaki et al., 2016*). Thus, BMP2 in the visceral endoderm in PreG1 embryo is thought not to be the trigger for the initial amniogenesis processes that forms the amniotic sac, and it is currently unclear how BMP signaling is activated in the nascent amnion cells during implantation in vivo. Using a Gel-3D hPSC-amnion system, we previously showed that amniogenesis is mechanosensitive, and that a soft substrate can induce BMP signaling in a mechanically dependent manner in developing hPSC aggregates/cysts. This suggests that mechanical signaling might play an important role in the initial BMP signaling activation and downstream amniogenesis during implantation in vivo. In contrast, since a continuous BMP signaling cascade is established in epiblast cells of the early primate gastrula (e.g. driving primitive streak formation, primordial germ cell formation), we speculate that the amniogenesis process that occurs during and post-gastrulation may be independent of a mechanical cue. It is, however, important to note that the pluripotency states are distinct in the epiblast cells during implantation compared to during gastrulation; it has been suggested that the epiblast cells are in the formative state during implantation and in the primed state during gastrulation (*Kinoshita and Smith, 2018*; *Shahbazi et al., 2017*; *Smith, 2017*). Thus, it is possible that the mechanisms regulating responses to BMP ligands as well as BMP-dependent transcriptional machineries may differ.

Overall, our detailed temporally resolved analyses of Glass-3D+BMP hPSC-amnion development, combined with validation in early macaque embryos, has revealed several previously unrecognized findings with implications for BMP-dependent amniogenic transcriptional machineries during amniogenesis, mechanisms continuing amniogenesis in peri-gastrula macaque embryos, and functional characterization of the transcriptional machinery required for amnion fate specification.

## Materials and methods
### hESC lines

Two human embryonic stem cell lines, H9 (WA09, P30, P48, WiCell; NIH registration number: 0062) and H7 (WA07, P50, WiCell; NIH registration number: 0061), were used in this study. All protocols for the use of hPSC lines were approved by the Human Stem Cell Research Oversight Committee at the Medical College of Wisconsin. All hPSC lines were maintained in a feeder-free system for at least 20 passages and authenticated as karyotypically normal at the indicated passage number. Karyotype analysis was performed at Cell Line Genetics. All hPSC lines tested negative for mycoplasma contamination (LookOut Mycoplasma PCR Detection Kit, Sigma-Aldrich). All transgenic and KO hESC lines in this study used H9 as the parental line. In summary, hESC were maintained in a feeder-free culture system with mTeSR1 medium, or with 50%/50% mix of mTeSR1 and mTeSR plus (STEMCELL Technologies). hESC were cultured on 1% (vol/vol) Geltrex (Thermo Fisher Scientific), or with Cultrex SCQ (Bio-Techne) coated six-well plates (Nunc). Cells were passaged as small clumps every 4–5 days with Dispase (Gibco). All cells were cultured at 37°C with 5% $CO_2$. Media was changed every day. hESC were visually checked every day to ensure the absence of spontaneously differentiated mesenchymal-like cells in the culture. Minor differentiated cells were scratched off the plate under a dissecting scope once identified. The quality of all hESC lines was periodically examined by immunostaining for pluripotency markers and successful differentiation to three germ layer cells.

## Cynomolgus macaque

Animals

The female and male cynomolgus macaque were housed and cared for at the Wisconsin National Primate Research Center (WNPRC). All procedures were performed in accordance with the NIH Guide for the Care and Use of Laboratory Animals and under approval of the University of Wisconsin College of Letters and Sciences and Vice Chancellor Office for Research and Graduate Education Institutional Animal Care and Use Committee (protocol g005061).

Animal breeding and pregnancy detection

Beginning on day 8 post-onset of menses, the female was housed with a compatible male and monitored for breeding. Animals were pair-housed until day 16–20 post-onset of menses. A 2–3 mL blood draw was performed daily from day 8 post-onset of menses until day 16 to assess the timing of ovulation based on the estradiol peak and rise in progesterone in serum. Serum samples were analyzed for estradiol (25 µL) and progesterone (20 µL) using a cobas e411 analyzer equipped with ElectroChemiLuminescence technology (Roche, Basal, Switzerland) according to the manufacturer's instructions. Results were determined via a calibration curve which was instrument-generated by 2-point calibration using traceable standards and a master curve provided via the reagent barcode. Inter-assay coefficient of variation (CV) was determined by assaying aliquots of a pool of rhesus plasma. For estradiol, the limit of quantitation (LOQ) was 25 pg/mL, the intra-assay CV was 2.02%, and the inter-assay CV was 5.05%. For progesterone, the LOQ was 0.2 ng/mL, the intra-assay CV was 1.37%, and the inter-assay CV was 4.63%. A transabdominal ultrasound was performed to detect pregnancy as early as 14 days post-ovulation. The ultrasound measurements in combination with the timing of ovulation were used to estimate the day of conception and gestational age of the pregnancy.

Terminal perfusion uterine collection, paraffin embedding, sectioning, and staining

The pregnant female was sedated with intramuscular ketamine (>15 mg/kg) followed by IV sodium pentobarbital (>35 mg/kg) and then perfused with 4% paraformaldehyde (PFA) via the left ventricle. The entire uterus and cervix were removed. The serosa and superficial myometrium were scored for better fixative penetration and to denote dorsal and ventral orientation. Tissues were fixed in 4% PFA with constant stirring and solution changes were performed every 24 hr for a total of 72 hr. The uterus was serially sectioned from the fundus to the internal cervical os into 4 mm slices using a dissection box. Cassettes with tissue sections were transferred into 70% ethanol, routinely processed in an automated tissue processor and embedded in paraffin for histological analyses (5 µm sections). Fixed uterine tissue were cut in 5 µm thick cross-sections, mounted on slides, deparaffinized in xylene and rehydrated in an ethanol series. Antigen retrieval was performed by boiling in citrate buffer. Sections were blocked 4% goat serum in phosphate-buffered saline (PBS) at room temperature for at least 3 hr. Subsequent immunolocalization was performed using commercially available primary antibodies, incubated overnight at 4°C in 4% serum, at the dilutions as shown in *Supplementary file 3*. Immunofluorescent detection was performed using secondary antibodies tagged with a fluorescent dye (fluorophores excitation = 488, 555, and 647 nm), and counterstained with DAPI. Negative controls were performed in which the primary antibody was substituted with the same concentration of normal IgG of the appropriate isotype. Images were obtained with a Zeiss LSM980 microscope.

## Glass-3D$^{+BMP}$ hPSC-amnion formation assays

Glass-3D protocol used to generate hPSC-pluripotent cysts was previously described (*Hamed et al., 2023*; *Shao et al., 2017a*; *Taniguchi et al., 2017*; *Taniguchi et al., 2015*). In brief, hPSC monolayers were dissociated using Accutase (Sigma-Aldrich) at 37°C for 10 min, centrifuged and resuspended in mTeSR1/mTeSR plus 50:50 mix containing 10 µM Y-27632 (STEMCELL Technologies). hPSC were plated as single cells at 20,000 cells/cm$^2$, unless otherwise specified, onto a 1% (vol/vol) gel coated wells (Geltrex [Thermo Fisher Scientific], or with Cultrex SCQ [Bio-Techne]). After 24 hr (on day 1), culture medium was replenished with fresh mTeSR1/mTeSR plus 50:50 media without Y-27632, supplemented with 2% (vol/vol) Geltrex or with Cultrex SCQ. Thereafter, media change was performed daily until day 4 (using mTeSR1/mTeSR plus 50:50 media supplemented with 1% [vol/vol] Geltrex or

with Cultrex SCQ), unless otherwise noted. This Glass-3D culture condition results in the formation of hPSC-cysts, cysts with a central lumen surrounded by pluripotent columnar cells. The Glass-3D$^{+BMP}$ culture condition was used to generate hPSC-amnion, cysts with a central lumen surrounded by squamous amnion-like cells. To induce amniogenesis, exogenous BMP ligand (20 ng/mL, Bio-techne) was added to developing pluripotent cysts between day 2 and 3 for 24 hr, or between day 2 and 4 for 48 hr, unless otherwise noted (e.g. for time-course analyses). Samples were harvested at day 4 (48 hr since starting BMP4 treatment). LDN-193189 (500 nM, STEMCELL Technologies) was used to disrupt BMP signaling.

## DNA constructs

### piggyBac-CRISPR/Cas9 (pBACON) constructs

A piggyBac-CRISPR/Cas9 (pBACON) vector that contains SpCas9-T2A-puro and hU6-gRNA expression cassettes flanked by piggyBac transposon terminal repeat elements (pBACON-puro), which allows subcloning of annealed oligos containing gRNA sequence at *BbsI* site, has been previously described (*Shao et al., 2017b*; *Townshend et al., 2020*; *Wang et al., 2021*). gRNA targeting genomic sites and oligo sequences to generate pBACON-GFP-h*GATA3* (primers: CRISPR_hGATA3_CDS1#1_s and CRISPR_hGATA3_CDS1#1_as) and -h*TFAP2A* (CRISPR_hTFAP2A_CDS2_s and CRISPR_hTFAP2A_CDS1_as) are found in *Figure 8—figure supplement 1* and *Supplementary file 3B*; these were designed using publicly available tools (available here or or here).

## piggyBac-based transgenic and genome edited hESC lines

PB constructs (3 µg) and pCAG-ePBase (*Lacoste et al., 2009*) (1 µg) were co-transfected into H9 hESC (70,000 cells/cm$^2$) using GeneJammer transfection reagent (Agilent Technologies). To enrich for successfully transfected cells, drug selection (puromycin, 2 µg/mL for 4 days; neomycin, 250 µg/mL for 5 days) was performed 48–72 hr after transfection. hESC stably expressing each construct maintained the expression of pluripotency markers. For inducible constructs, DOX (500 ng/mL) treatment was performed for 48 hr for neomycin-selected *TFAP2A*-KO rescue pools using PB-TA-ERN-hTFAP2A(CR)-HA. The TFAP2A gRNA targeted site of human TFAP2A construct (gift of Robert Tjian, Addgene #12100, *Williams and Tjian, 1991*) was mutated using Q5 site directed mutagenesis kit (NEB, primers: SP (RSV) AP2 (hTFAP2A)-CRISPR RES-fw and SP (RSV) AP2 (hTFAP2A)-CRISPR RES-rv). This CRISPR resistant human TFAP2A construct was then PCR amplified (using primers: Clo-dTOPO-hTFAP2A-CRISPR res-fw v2 and Clo-dTOPO-hTFAP2A-CRISPR res-rv v2, HA tagging at C-terminus), which was then cloned into pENTR-dTOPO (Thermo) for Gateway cloning into PB-TA-ERN (Knut Woltjen, Addgene #80474, *Kim et al., 2016*) to generate PB-TA-ERN-hTFAP2A(CR)-HA. Additional plasmid and primer information is found in *Supplementary file 3*.

During pBACON-based genome editing, puro-selected cells were cultured at low density (300 cells/cm$^2$) for clonal selection. Established colonies were manually picked and expanded for screening indel mutations using PCR amplification of a region spanning the targeted gRNA region (genomic DNA isolated using DirectPCR Lysis Reagent [Tail] [VIAGEN], primers: hGATA3, PCR_GATA3_RI_fw#2 and PCR_GATA3_NI_rv#2; hTFAP2A, PCR_TFAP2A_RI_fw and PCR_TFAP2A_NI_rv), which were subcloned into pPBCAG-GFP (*Chen and LoTurco, 2012*) at EcoRI and NotI sites, and sequenced (Seq-3'TR-pPB-Fw). At least 12–15 bacterial colonies were sequenced to confirm genotypic clonality. Western blots for each protein were also performed to further validate the KO lines. Control cells are H9 hESC in all loss-of-function experiments.

## Western blot

SDS-PAGE gels (10%) and PVDF membranes were used. Membranes were blocked using Intercept (TBS) Blocking Buffer (Licor), total protein quantification was performed by using Revert 700 Total Protein Stain (Licor), and primary antibody overnight incubation was performed at 4°C, followed by 30 min IRDye (Licor) secondary antibody incubation at room temperature. Blots were imaged using Licor Odyssey Infrared Imaging system.

## Immunostaining

hPSC staining protocols have been previously described (e.g., *Hamed et al., 2023*; *Shao et al., 2017a*; *Wang et al., 2021*). In short, samples (hPSC-cysts, hPSC-amnion) grown on the coverslip were

rinsed with PBS (Gibco) twice, fixed with 4% PFA (Sigma-Aldrich) in 1× PBS for 40–60 min, then rinsed with PBS three times, and permeabilized with 0.1% SDS (Sigma-Aldrich) solution for 40 min. The samples were blocked in 4% heat-inactivated goat serum (Gibco) or 4% normal donkey serum (Gibco) in PBS 1 hr to overnight at 4°C. The samples were incubated with primary antibody solution prepared in blocking solution at 4°C overnight, washed three times with PBS (10 min each), and incubated in blocking solution with goat or donkey raised Alexa Fluor-conjugated secondary antibodies (Thermo Fisher Scientific) at room temperature for 2 hr. DNA counterstaining was performed using Hoechst 33342 (nucleus, Thermo Fisher Scientific), Alexa Fluor-conjugated WGA (membrane, Thermo Fisher Scientific). All samples were mounted on slides using 90% glycerol (in 1× PBS). When mounting cyst samples, modeling clays were used as spacers between coverslips and slides to preserve lumenal cyst morphology.

## Confocal microscopy, live imaging, and image analysis

Confocal images of fixed samples were acquired using Zeiss LSM980 microscope. Measurements of cell nucleus dimensions, cyst epithelial thickness, and cyst orientation angle were performed manually using the Measurement Tool in ImageJ (NIH).

## RNA isolation and qRT-PCR analysis

Total RNA extraction was performed using TRIzol (Thermo Fisher). The quantity and quality of total RNA was determined by spectrometry and denaturing agarose gel electrophoresis, respectively. The cDNA was synthesized from total RNA (1 µg) using High-Capacity cDNA Reverse Transcription Kit (Applied Biosystems). Quantitative real-time RT-PCR (qPCR) analysis of mRNA expression was performed using a CFX96 Real-Time PCR Detection System (Bio-Rad) with Power Up SYBR Green Master Mix (Applied Biosystems). Primers (*Supplementary file 3B*) were designed using NCBI primer design tool to span at least one exon/intron junction. Data were normalized against *GAPDH* and are shown as the average fold increase ± standard error of the mean (SEM) with relative expression being calculated using the 2-ΔΔCT method as described previously (*Livak and Schmittgen, 2001*). After amplification, the specificity of the PCR was determined by both melt-curve analysis and gel electrophoresis to verify that only a single product of the correct size was present.

## RNA sequencing and bioinformatics

The quality of total RNA was determined using TapeStation RNA ScreenTape. RNA was enriched for mRNA using NEBNext Poly(A) mRNA Magnetic Isolation Module (E7490). cDNA and library preparation were performed using NEBNext Ultra II Directional RNA Library Prep kit for Illumina (New England Biolabs). Library concentration was determined using NEBNext Library Quant Kit for Illumina (New England Biolabs) or by Qubit (Thermo). Samples were then pooled and were sequenced using Illumina sequencers: NextSeq550 (Versiti Blood Research Institute) or using NovaSeq 6000 (S4 flow-cell, the DNA services laboratory of the Roy J. Carver Biotechnology Center at the University of Illinois at Urbana-Champaign). Sequence data was analyzed using the RNA sequencing workflow in Partek Flow software (Partek Inc, St. Louis, MO, USA). Raw sequencing reads were mapped to the human genome hg38 using STAR v.2.7.8a with default parameter settings. Aligned reads were then quantified to Ensembl Transcripts release 104 using Partek's method and genes with fewer than 10 reads in all samples were excluded from the subsequent analyses. Gene counts were then filtered based on gene_biotype for protein_coding genes, followed by excluding features where maximum ≤ 30.0 reads. Gene expression was then normalized using DESeq2 (Median Ratio). Differentially expressed genes were selected using as criteria FDR<0.05 and absolute fold change >2.

### GO enrichment analysis

GO analysis was performed on significantly upregulated (1820) or downregulated (2092) genes in Glass-3D⁺ᴮᴹᴾ amnion-cysts vs Glass-3D pluripotent-cysts (from *Figure 3*).

### Mapping and gene expression analyses of Tyser et al. CS7 human embryo single-cell RNA sequencing dataset

The original UMAP coordinates and annotations for all 1195 cells were used throughout. The processed data were downloaded from http://www.human-gastrula.net, which was used to generate a UMAP

plot (using DimPlot in R package Seurat), as well as to perform expression analyses (FeaturePlot in Seurat).

## Integration and gene expression analyses of Yang et al. GD14 cynomolgus macaque embryo single-cell RNA sequencing dataset

The processed data were downloaded from GSE148683, and the ensemble genome build Macaca_fascicularis_5.0 release 96 was used to identify human orthogonal gene symbols. Datasets from two distinct GD14 embryos (*Yang et al., 2021*) were integrated using 5000 anchor features and 10 dimensions in the IntegrateData function in Seurat R package (v.4.2.1) package: trophoblast cells were removed from the dataset prior to integration. Six general cell populations (epiblast, transition, mesoderm, amnion, endoderm, and extraembryonic mesenchyme) were identified using FindClusters function (resolution as 0.4).

## RNA velocity (scVelo) analysis of Tyser et al. CS7 human embryo single-cell RNA sequencing dataset

RNA velocities across all 1195 single cells were embedded on the original UMAP coordinates shown in *Tyser et al., 2021*. The Tyser et al. CS7 human embryo raw fastq dataset was obtained from Array-Express (accession code: E-MTAB-9388). From the aligned BAM files, a loom file was generated using the function run_smartseq2 mode in software velocyto (v.0.17) with default parameters to create a count matrix made of spliced and unspliced read counts (with human genome annotation hg19). scVelo (Python package v.0.2.5, *Bergen et al., 2020*) was used to examine inferred RNA velocity and visualized the velocities on the UMAP embedding by stream plot (with default parameters except smooth was set to 1).

## Transcriptomic similarity analysis between bulk RNA sequencing and Tyser et al. CS7 human embryo or Yang et al. GD14 cynomolgus macaque peri-gastrula single-cell RNA sequencing datasets

Transcriptomic similarities between bulk RNA sequencing samples and the Tyser et al. dataset were examined and visualized using original UMAP coordinates from Tyser et al. In summary, each bulk RNA sequencing sample was first downsampled to 100 pseudo-single-cell data using R package projectLSI (*Granja et al., 2019*), which was followed by data integration with Tyser et al. dataset in UMAP dimension reduction space. A medoid from 100 pseudo-single-cell data was used as the representation of each bulk RNA sequencing sample, and, for each sample, we identified the most transcriptomically similar cell from Tyser et al. dataset, based on Euclidean distance on UMAP space. Identical approaches were used for *Yang et al., 2021*, except that the UMAP coordinates were identified using the Seurat package as described above.

### Statistical analyses

Graphs were generated using Prism 7 (GraphPad Software) or R, and statistical analyses were performed using Prism 7. *Figure 1D*, five independent samples were counted at each timepoint (total of >50 cells/sample). For analysis we used one-way ANOVA with post hoc Dunnet's multiple comparisons test. In *Figure 1K*, three independent samples were counted (total of >75 cells were plotted). For analysis we used two-tailed Student's t-test. In *Figure 1J*, three independent samples were counted (total of >50 cells were plotted). For analysis we used one-way ANOVA with post hoc Tukey's multiple comparison test. In *Figure 2*, relative mRNA expression was detected in n=3 for each timepoint. For analysis we used one-way ANOVA with post hoc Tukey's multiple comparison test. *, significant (p-value<0.05); ns, non-significant. All experiments were repeated at least three times.

### Acknowledgements

We thank K Woltjen (Kyoto University) for the piggyBac DOX-inducible vector (PB-TA-ERN, Addgene#80474), A Brivanlou (Rockefeller University) for ePB transposase, as well as Robert Tjian

(University of California – Berkley) for human TFAP2A plasmid (SP (RSV) AP-2, Addgene #12100). We thank Michaela Patterson, Matthew Veldman, and Deborah Gumucio for their insightful comments to the manuscript, as well as Versiti Blood Research Institute Molecular Biology Core Lab and the DNA Services laboratory of the Roy J Carver Biotechnology Center at the University of Illinois at Urbana-Champaign for sequence services. We also thank the WNPRC Veterinary, Scientific Protocol Implementation, Pathology and Animal Services Unit staff for providing animal care, and assisting in procedures including breeding, pregnancy monitoring, and sample collection. The contents of this manuscript are solely the responsibility of the authors and do not represent the official views of the NIH.

## Additional information

### Funding

| Funder | Grant reference number | Author |
|---|---|---|
| Eunice Kennedy Shriver National Institute of Child Health and Human Development | R01-HD098231 | Kenichiro Taniguchi |
| National Institutes of Health | P51-OD011106 | Thaddeus G Golos |
| Medical College of Wisconsin | CBNA Start-up funds | Kenichiro Taniguchi |
| Advancing a Healthier Wisconsin (AHW) Endowment | 16003-5520766 | Nikola Sekulovski |
| The Lalor Foundation Postdoctoral Fellowship | | Nikola Sekulovski |

The funders had no role in study design, data collection and interpretation, or the decision to submit the work for publication.

### Author contributions

Nikola Sekulovski, Conceptualization, Data curation, Formal analysis, Validation, Investigation, Visualization, Methodology, Writing - original draft, Writing – review and editing; Jenna C Wettstein, Linnea E Taniguchi, Xiaolong Ma, Investigation; Amber E Carleton, Lauren N Juga, Investigation, Writing – review and editing; Sridhar Rao, Resources, Investigation, Writing – review and editing; Jenna K Schmidt, Conceptualization, Resources, Investigation, Methodology, Project administration, Writing – review and editing; Thaddeus G Golos, Conceptualization, Resources, Formal analysis, Supervision, Funding acquisition, Investigation, Methodology, Project administration, Writing – review and editing; Chien-Wei Lin, Conceptualization, Data curation, Software, Formal analysis, Supervision, Methodology, Writing – review and editing; Kenichiro Taniguchi, Conceptualization, Resources, Data curation, Software, Formal analysis, Supervision, Funding acquisition, Validation, Investigation, Visualization, Methodology, Writing - original draft, Project administration, Writing – review and editing

### Author ORCIDs

Nikola Sekulovski ⑩ https://orcid.org/0000-0003-3451-7262
Amber E Carleton ⑩ https://orcid.org/0000-0002-7347-6945
Xiaolong Ma ⑩ https://orcid.org/0009-0009-9196-182X
Chien-Wei Lin ⑩ https://orcid.org/0000-0003-4023-7339
Kenichiro Taniguchi ⑩ https://orcid.org/0000-0002-4531-7197

### Ethics

The female and male cynomolgus macaque were housed and cared for at the Wisconsin National Primate Research Center (WNPRC). All procedures were performed in accordance with the NIH Guide for the Care and Use of Laboratory Animals and under approval of the University of Wisconsin College

of Letters and Sciences and Vice Chancellor Office for Research and Graduate Education Institutional Animal Care and Use Committee (protocol g005061).

Reviewer #2 (Public review): https://doi.org/10.7554/eLife.89367.3.sa1
Reviewer #3 (Public review): https://doi.org/10.7554/eLife.89367.3.sa2
Author response https://doi.org/10.7554/eLife.89367.3.sa3

## Additional files

### Supplementary files
• Supplementary file 1. Glass-3D$^{+BMP}$ human pluripotent stem cell (hPSC)-amnion time-course bulk RNA sequencing dataset. (A) Normalized and filtered count matrix of the time-course bulk-RNA sequencing. (B–C) Gene expression comparison analysis between 48 hr (d4) Glass-3D$^{+BMP}$ hPSC-amnion and d4 Glass-3D hPSC-cysts: lists of 1813 upregulated (B) and 2088 down- (C) regulated genes. (D–E) Gene Ontology analyses (Biological Processes) for top all significantly up- (D) and down- (E) regulated genes.

• Supplementary file 2. Lists of immediate, early, intermediate, and late genes. An excel file containing temporally resolved gene lists: (A) immediate (0.5 hr and 1 hr, 57 genes); (B) early (3 hr and 6 hr, 135 genes); (C) intermediate-1 (12 hr, 81 genes); (D) intermediate-2 (24 hr, 642 genes); (E) late (48 hr, 909 genes). Genes that showed significant upregulation compared to the previous timepoint (e.g. 0.5 hr compared to 0 hr, 3 hr compared to 1 hr) were retained. Duplicated genes found in the list from earlier timepoint lists are removed. For example, *GATA3* is significantly upregulated throughout the time-course, but is only included in the immediate category. Genes with averaged normalized counts <100 were excluded.

• Supplementary file 3. Lists of reagents used in this study. (A–C) Lists of antibodies (A), primers (B), and plasmids (C).

• MDAR checklist

### Data availability
Time-course bulk RNA sequencing data have been deposited under controlled access in the database of Genotypes and Phenotypes (dbGaP) under accession phs002184.v1.p1 (in line with NIH grant (R01-HD098231) requirement for human subject sequencing data). Only senior investigators can request access to the dbGaP data and a Research Use Statement (similar to proposal) needs to be submitted; the NICHD Data Access Committee will review the data access requests (HD-DAC@mail.nih.gov). Normalized and filtered count matrix of the time-course bulk RNA sequencing data is available in Supplementary file 1A.

The following dataset was generated:

| Author(s) | Year | Dataset title | Dataset URL | Database and Identifier |
|---|---|---|---|---|
| Sekulovski N, Lin CW, Taniguchi K | 2024 | Transcriptomic analysis of pluripotent stem cell-based model of human amniogenesis | https://www.ncbi.nlm.nih.gov/projects/gap/cgi-bin/study.cgi?study_id=phs002184.v1.p1 | NCBI dbGaP, phs002184.v1.p1 |

The following previously published datasets were used:

| Author(s) | Year | Dataset title | Dataset URL | Database and Identifier |
|---|---|---|---|---|
| Tyler RCV, Mahammadov E, Nakanoh S, Vallier L, Scialdone A, Srinivas S | 2021 | A spatially resolved single cell atlas of human gastrulation | https://www.ebi.ac.uk/biostudies/arrayexpress/studies/E-MTAB-9388 | ArrayExpress, E-MTAB-9388 |

*Continued on next page*

*Continued*

| Author(s) | Year | Dataset title | Dataset URL | Database and Identifier |
|---|---|---|---|---|
| Yang R, Goedel A, Kang Y, Si C, Chu C, Zheng Y, Chen Z, Gruber PJ, Xiao Y, Zhou C, Witman N, Eroglu E, Leung CY, Chen Y, Fu J, Ji W, Lanner F, Niu Y, Chien KR | 2021 | Amnion signals are essential for mesoderm formation in primates | https://www.ncbi.nlm.nih.gov/geo/query/acc.cgi?acc=GSE148683 | NCBI Gene Expression Omnibus, GSE148683 |

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
