## [Editor Report · eLife assessment]

This study presents an **important** dataset that captures the transition from epiblast to amnion using a novel in vitro model of human amnion formation. The supporting evidence for the authors' claims is **convincing**. Key strengths of the study include the efficiency and purity of the cell populations produced, a high degree of synchrony in the differentiation process, comprehensive benchmarking with single-cell data and immunocytochemistry from primate embryos, and the identification of critical markers for specific differentiation phases. A notable limitation, however, is the model's exclusion of other embryonic tissues.

---

## [Referee Report · Reviewer #2 (Public review)]

In this study, Sekulovski and colleagues report refinements to an in vitro model of human amnion formation. Working with 3D cultures and BMP4 to induce differentiation, the authors chart the time course of amnion induction in human pluripotent stem cells in their system using immunofluorescence and RNA-seq. They carry out validation through comparison of their data to existing embryo datasets, and through immunostaining of post-implantation marmoset embryos. Functional experiments show that the transcription factor TFAP2C drives the amnion differentiation program once it has been initiated.

There is currently great interest in the development of in vitro models of human embryonic development. While it is known that the amnion plays an important structural supporting role for the embryo, its other functions, such as morphogen production and differentiation potential, are not fully understood. Since a number of aspects of amnion development are specific to primates, models of amniogenesis will be valuable for the study of human development. Advantages of this model include its efficiency and the purity of the cell populations produced, a significant degree of synchrony in the differentiation process, benchmarking with single-cell data and immunocytochemistry from primate embryos, and identification of key markers of specific phases of differentiation. Weaknesses are the absence of other embryonic tissues in the model, and overinterpretation of certain findings, in particular relating bulk RNA-seq results to scRNA-seq data from published analyses of primate embryos and results from limited (though high quality) embryo immunostainings.

---

## [Referee Report · Reviewer #3 (Public review)]

In this work, the authors tried to profile time-dependent changes in gene and protein expression during BMP-induced amnion differentiation from hPSCs. The authors depicted a GATA3 - TFAP2A - ISL1/HAND1 order of amniotic gene activation, which provides a more detailed temporary trajectory of amnion differentiation compared to previous works. As a primary goal of this study, the above temporal gene/protein activation order is amply supported by experimental data. However, the mechanistic insights on amniotic fate decision, as well as the transcriptomic analysis comparing amnion-like cells from this work and other works remain limited. While this work allows us to see more details of amnion differentiation and understand how different transcription factors were turned on in a sequence and might be useful for benchmarking the identity of amnion in ex utero cultured human embryos/embryoids, it provides limited insights on how amnion cells might diverge from primitive streak / mesoderm-like cells, despite some transcriptional similarity they shared, during early development.

[Editors' note: In the revised manuscript, the authors have added new results and made textual revisions that address the reviewers' concerns. These changes have significantly enhanced the clarity, quality, and impact of the study.]

---

## [Author Response]

The following is the authors’ response to the original reviews.

We appreciate the reviewers for their insightful comments, which have helped to improve the manuscript. We provide specific examples and a point-by-point response to all comments, below. Based on the Reviewers’ comments, we revised our manuscript, adding considerable amount of new data (found in Fig. 1A,B, 4E-G, 7C,D, 8C,E, S1B,C, S2C-G, S4C, and Video 1). In the main manuscript text, blue fonts indicate added or revised texts. An additional author (Lauren N. Juga) is added for the newly generated data in the revised manuscript.

**Reviewer #1:**
Sekulovski et al present an interesting and timely manuscript describing the temporal transition from epiblast to amnion. The manuscript builds on their previous work describing this process using stem cell models.They suggest a multi-step process initiated by BMP induction of GATA3, followed by expression of TFAP2A, followed by ISL1/HAND1 in parallel with loss of pluripotency markers. This transition was reproduced through IF analysis of CS6/7 NHP embryo.There are significant similarities in the expression of trophectoderm and the amnion. There are also ample manuscripts showing trophoblast induction following BMP stimulation of primed pluripotent stem cells. The authors should ensure that the amnion indeed is only amnion and not trophectoderm (or the amount of contribution to trophectoderm). As an extension, does the amnion character remain after the 48h BMP4 treatment, and is a trophectoderm-like state adopted as suggested by Ohgushi et al 2022?

Thank you for this insightful comment. As pointed out, Ohgushi et al. showed that, in their culture method, amnion is first induced, and extended culturing leads to the formation of trophectoderm-like cells (Ohgushi et al., 2022).

Importantly, we would like to note that our culture system differs substantially from that of Ohgushi et al. in several respects. First our system uses a 3D culture method while Ohgushi et al. employ 2D hPSC monolayers. Second, the two systems are chemically quite distinct. In our Glass-3D+BMP protocol, cells are cultured in mTeSR media (which contains FGF2 and TGFb1) for two days, by which time they generate 3D pluripotent cysts. BMP is then added to the culture medium for 24 hours, followed by another 24 hours without BMP4. In stark contrast, Ohgushi et al. employ A83-01, an Activin/Nodal signaling inhibitor, and PD173074, an FGF signaling inhibitor (a protocol which they call AP). This treatment leads to spontaneous activation of BMP signaling, but it also clearly inhibits Activin/Nodal and FGF signaling pathways, which remain active in our system. As a result of these distinct chemical as well as geometrical culturing protocols, their system produces amnion and trophectoderm, while our system produces exclusively amnion.

Further analysis of gene expression data provides additional data supporting our contention that our system produces amnion. Though the gene expression profiles of amnion and trophectoderm are quite similar, specific markers of trophectoderm have been identified including GCM1, PSG1, PSG4 and CGB (Blakeley et al., 2015; Meistermann et al., 2021; Ohgushi et al., 2022; Okae et al., 2018; Petropoulos et al., 2016; Yabe et al., 2016). Importantly, while all of these markers are abundantly expressed in the Ohgushi et al. system, bulk RNA sequencing analysis of our Glass-3D+BMP hPSC-amnion cells reveals that none of these markers are detectable. Indeed, SDC1, a marker that Ohgushi et al. claim distinguishes trophoblast from amnion actually decreases (more than 8-fold) as pluripotent cysts transition to amnion in Glass3D+BMP. Finally, Ohgushi et al. report that ISL1, a key marker of specified amnion population, is initially increased in their system, but is reduced to a basal level overtime. In contrast, in Glass3D+BMP hPSC-amnion, ISL1 expression continuously increases with time, and ISL1 protein expression is seen uniformly throughout the amnion cysts. This uniform expression is also seen in CS6/7 cynomolgus macaque amnion. Together, these results support out conclusion that the Glass-3D+BMP system leads to the formation of amniotic cells, and not trophectoderm cells.

The functional data does not support a direct function of GATA3 prior to TFAP2A and the authors suggest compensatory mechanisms from other GATAs. If so, which GATAs are expressed in this system, with and without GATA3 targeting? Would it not be equally likely that the other early genes could be the key drivers of amnion initiation, such as ID2?

We appreciate this helpful comment. We agree that our data do not provide sufficient evidence for the role of GATA3 in early amniogenesis. We also agree that other early genes could be key drivers, and apologize for including our speculation that focuses only on GATA2. GATA2 was selected because, among the other GATAs, GATA2 and GATA3 are the only abundantly expressed GATA factors. This point suggesting a potentially redundant role of GATA2 is now removed from the manuscript (Line#355 of the original manuscript).

The targeting of TFAP2A displays a very interesting phenotype which suggests that amnion and streak share an initial trajectory but where TFAP2A is necessary to adopt amnion fate. It would again be important to ensure that this alternative fate is indeed in streak and not misannotated alternative lineages, including trophoblast.Is TBXT induced in this setting as well as in the wt situation during amnion induction? This should be displayed as in Figure 3D and would be nice to be complimented by NHP IF analysis.

We will address these two closely related comments together.

TFAP2A-KO cysts contain ISL1+ squamous cells as well as SOX2+ pluripotent cells, suggesting that, while the initial focal amniogenesis is seen, subsequent spreading event is not seen. Interestingly, our new data show that TFAP2A-KO cysts display cells with high TBXT expression (Fig. 8E, Line#373-374). This result suggests that, in the absence of TFAP2A, once amnion lineage progression is halted, more primitive streak-like (TBXThigh) lineage emerges. It is important to note that TBXT expression is not seen in the trophectoderm population of cynomolgus macaque peri-gastrula (Sasaki et al., 2016; Yang et al., 2021).

As suggested, we now include a TBXT expression time course during hPSC-amnion formation in Fig. S2D of the revised manuscript. These data show weak TBXT expression (transcripts) starting at the 24-hr timepoint. However, a clear TBXT protein signal could not be detected using IF (Fig. S2C), likely because TBXT expression is very low (Line#264-265). While statistically significant compared to the 12-hr timepoint, TBXT expression is 31 FPKM +/- 0.8 (standard deviation) at 24-hr and 48 FPKM +/- 6 at 48-hr. These are low expression values compared to, for example, TFAP2A, which displays 572 FPKM +/- 23 at 12-hr and 1169 FPKM +/- 27 at 24-hr, at which TFAP2A is readily detected using IF. While weak nuclear TFAP2A is seen using IF at 6hr (187 FPKM +/- 7), no clear TFAP2A is detected at 3-hr (74 FPKM +/- 7). Another example is ISL1, which displays 758 FPKM +/- 55 at 24-hr and 1505 FPKM +/- 26 at 48-hr, when ISL can be detected using IF. Importantly, we were not able to detect ISL1 protein expression using IF at

12-hr, at which its expression level is 12 FPKM +/-18. Lastly, we now show that, in the cynomolgus macaque peri-gastrula, while pSMAD1/5+ primitive streak-derived disseminating cells show abundant TBXT expression, no clear TBXT expression is seen in the amnion territory (Fig. S2G, Line#291-293).

Together, these results show that while a TBXTlow state clearly emerges during hPSC-amnion development, in wild-type hPSC cultured in Glass-3D+BMP, TBXT levels remain low throughout amnion differentiation. However, in the absence of TFAP2A, a TBXThigh state is seen, suggesting that TFAP2A is critical for suppressing this TBXThigh state in fate spreading cells, perhaps by preventing BMP responding cells from acquiring embryonic lineages (e.g., mesodermal and/or primordial germ cells).

The authors should address why they get different results from Castillo-Venzor et al 2023 DOI: 10.26508/lsa.202201706

Thank you very much for this helpful suggestion, and we now include a section detailing this in the Discussion (Line#410-432). In short, we propose several possibilities. First, culturing conditions are highly distinct. Castillo-Venzor et al. (Castillo-Venzor et al., 2023) utilize initial “pre-mesoderm” conditioning by Activin and CHIR, followed by treating floating embryoid bodies with a growth factor cocktail (BMP, SCF, EGF and LIF). In contrast, our system (Glass-3D+BMP) employs BMP stimulation of pluripotent cysts. Thus, we suspect that, in the PGCLC differentiation condition, cells are conditioned to the pre-mesodermal lineage. Moreover, we propose that amnion fate spreading may not be present in the PGCLC system, perhaps due to differences in geometry (aggregates versus cysts), or due to differing lineage commitment programs. That is, while initial amniogenesis is seen in the PGCLC system, most cells may already be committed to the PGC-like or mesodermal lineages by the time amnion fate spreading can occur. Alternatively, because several cell types (PGC-like, mesodermal and amniotic) co-exist in the culture by Castillo-Venzor et al., PGC-like and/or mesodermal cells may compensate for the loss of TFAP2A.

**Reviewer #2:**
In this study, Sekulovski and colleagues report refinements to an in vitro model of human amnion formation. Working with 3D cultures and BMP4 to induce differentiation, the authors chart the time course of amnion induction in human pluripotent stem cells in their system using immunofluorescence and RNA-seq. They carry out validation through comparison of their data to existing embryo datasets, and through immunostaining of post-implantation marmoset embryos. Functional experiments show that the transcription factor TFAP2C drives the amnion differentiation program once it has been initiated.There is currently great interest in the development of in vitro models of human embryonic development. While it is known that the amnion plays an important structural supporting role for the embryo, its other functions, such as morphogen production and differentiation potential, are not fully understood. Since a number of aspects of amnion development are specific to primates, models of amniogenesis will be valuable for the study of human development. Advantages of this model include its efficiency and the purity of the cell populations produced, a significant degree of synchrony in the differentiation process, benchmarking with single-cell data and immunocytochemistry from primate embryos, and identification of key markers of specific phases of differentiation. Weaknesses are the absence of other embryonic tissues in the model, and overinterpretation of certain findings, in particular relating bulk RNA-seq results to scRNA-seq data from published analyses of primate embryos and results from limited (though high quality) embryo immunostainings.

We are happy that Reviewer #2 agrees that our Glass-3D+BMP model is important for investigating additional roles of amniogenesis, as well as roles of amnion as a signaling hub, due to the purity of the amniotic cell population, and a high degree of synchrony of differentiation.

We respectfully disagree that the absence of other embryonic tissues in the model is a weakness: rather, we believe it is a strength because this single lineage amnion model allows us to directly (and independently) investigate mechanisms underlying amnion lineage progression. For example, as noted above in our response to Reviewer #1, use of our hPSCamnion model allowed us to see a very specific and interesting phenotype in the absence of TFAP2A (reduced amnion formation and emergence of an alternative lineage), though previous findings by Castilllo-Venzor et al. concluded that amniogenesis is not affected by loss of TFAP2A. We noted that the culture method used by Castillo-Venzor et al. contains several cell types (amniotic, mesodermal and PGC-like), and that amniogenesis may be intact in that model due to compensation by the presence of these other cell types. That is, while cell-cell interactions can indeed be gleaned in culture systems with several cell types, the presence of multiple cell types and their additional signaling inputs can also confound some aspects of mechanistic investigations. We now include a paragraph in the Discussion of the revised manuscript (Line#410-432), in which we detail these ideas, and suggest that, because of the cell purity, our Glass-3D+BMP model enables robust mechanistic examinations, specifically during amnion formation.

We address Reviewer #2’s point about bulk vs. single cell transcriptomic similarity analysis in Reviewer’s specific point #4 below. We do, however, want to note here that we have performed the same analysis using a 14-day old cynomolgus macaque peri-gastrula single cell RNA sequencing dataset generated by Yang et al. (Yang et al., 2021), and obtained a lineage trajectory (Fig. 4F, Line#265-268) similar to that seen when the Tyser et al. dataset (Tyser et al., 2021) was used (Fig. 4C).

Importantly, while cynomolgus macaque early embryo samples are limited, we now include additional staining (Fig. S2G).

**Reviewer #2 (Recommendations For The Authors):**
Provide more confirmation of key findings in more than one stem cell line.

We now confirm key findings in the H7 human embryonic stem cell line (Fig. S1C).

Provide stronger evidence e.g. scRNA-seq to support the existence of intermediate cells or tone down the conclusions.

We agree that this is a very important point. In our recent study (Sekulovski et al., 2023), we performed single cell RNA sequencing of Gel-3D, another hPSC-amnion model. In this study, we comprehensively described the transcriptome associated with the “intermediate” cell types, as well as CLDN10 as a marker of these cell types. Moreover, we now include additional data showing the molecular characteristics of the TBXTlow intermediate cells during amniogenesis in hPSC-amnion (Fig. S2C, S2D) and d14 cynomolgus macaque peri-gastrula (Fig 4G, replot of single cell RNAseq by (Yang et al., 2021), Line#264-268).

Provide more data on the expression of DLX5 in the model.

We now provide a DLX5 staining time course in Fig. 7C. We find that, similar to ISL1, prominent DLX5 staining is seen in the focal cells at 24-hr post-BMP. Interestingly, at 48-hr, while some cells show high levels of DLX5, some cells show low DLX5 levels; this is of an interest for future investigations.

(1) L159 - the authors should repeat more of the key results in at least one other hPSC line, to ensure reproducibility of the method. Figure S1 contains minimal information (one timepoint, three genes, one biological replicate) on a single different hPSC line.

We now include additional validation analysis using the H7 human ESC line (Fig. S1).

(2) Figure 1- it is a little difficult to appreciate cyst formation from images taken at one level in the stack, can the authors perhaps show a 3D rendering or video to display morphogenesis better?

We now provide all optical sections of cysts shown in Movie 1.

(3) Figure 1-did the authors carry out podocalyxin staining? This is a standard marker for lumenogenesis.

We now provide PODXL staining (Fig. 1A,1B).

(4) L248 onwards and Figure 4-I am a little skeptical concerning conclusions drawn from an overlay of bulk RNA-seq onto scRNA-seq UMAP plots. I think the authors need to provide some strong justification for this approach. I would be particularly careful about concluding that cells depicted in Fig 4D represent an intermediate close to primitive streak and even more careful about claiming any lineage relationship between T-positive "primitive streak like intermediates" and the trajectory of cells in the model. UMAP is a dimension-reduction technique for the visualization of clusters in high-dimensional data. It is not a lineage-tracing methodology. It would have been preferable for the authors to present their own scRNA-seq data from the model.

We are sorry that it was not clear that our approach to find similarity between bulk and single cell RNA-seq data is largely based on a published work (Granja et al., Nature Biotechnology 2019, (Granja et al., 2019)) named projectLSI. Please refer to our Methods section for details of the implementation and how we modified it for better visualization (addressed in Line#667-676 of the original manuscript, now in Line#718-730). The performance of projectLSI was extensively evaluated in the original article. Furthermore, as pointed out, UMAP is indeed a dimension reduction method that has been widely used in single cell RNA-seq research. In addition to visualizing clusters, trajectory analysis, such as RNA-velocity (which is used in this study), is another successful and widely adapted application of UMAP to gauge fate progression. Therefore, we believe that UMAP can be effectively used as a lineage prediction methodology, and that our use of bulk to single cell transcriptomic similarity analysis leveraging projectLSI is well justified at conceptual and technical levels.

As illustrated in Fig. 5A, we performed RNA-velocity analysis of the Tyser et al. dataset, and our result clearly predicts a differentiation trajectory from Epiblast, a part of the TBXTlow population shown in Fig. 4D, and, then, to Ectoderm/Amnion cells. Consistent with this bioinformatic result, we now show that some cells show some but weak TBXT expression (at the transcript level) at the 24-hr post-BMP timepoint in control hPSC-amnion (Fig. S2D, Line#264-265). Importantly, our conclusion is drawn from a trajectory based on our time course (0, 0.5, 1, 3, 6, 12, 24, and 48 hours post-BMP treatment) which shows a clear transition from epiblast cells to TBXTlow and then finally to the ectoderm/amnion population. Moreover, using the transcriptomic similarity analysis, we found that the loss of TFAP2A leads to emergence of more primitive streak-like transcriptional characteristics (Fig. 8D). Indeed, using IF, we now show that several fate spreading cells in the TFAP2A-KO cysts are TBXThigh (Fig. 8E, Line#373-374). Thus, the new data provide additional evidence for the successful implementation of this bulk/single cell transcriptomic similarity analysis.

Together, our bioinformatic and localization analyses show that the Glass-3D+BMP system recapitulates the trajectory found in our Tyser et al. RNA-velocity analysis, further supporting the validity of this differentiation trajectory. To avoid confusion, however, we now omit the “primitive streak-like” phrase when describing the TBXTlow cells because, while they may show some TBXT expression, they are likely intermediate fate transitioning cells. Indeed, a recent study by Ton et al. (Ton et al., 2023) showed that the Tyser et al. Primitive Streak cells consist of a mix of several lineage progressing cells (e.g., Epiblast, Non-neural ectoderm, Anterior or caudal primitive streak, PGC). Therefore, these cells are now specifically described as “TBXTlow” state; TBXThigh cells are described as primitive streak-like state.

(5) L276 Tyser data do come from a primate model; the authors mean NHP.

We now specifically state that the validation is performed in a non-human primate model (Line#280).

(6) Figure 5-though the immunostaining of the CS6/7 monkey embryos is excellent, the authors should not overinterpret these images. What is shown is not a time course, and one can only infer that a particular pattern of gene expression exists in a spatial sense from these images. In the model (Figure 2), the epiblast markers gradually fade and overlap for a time with emergent amnion markers, but in Figure 5 the transition between epiblast and amnion in the embryo seems pretty sharp, at least in terms of gene expression. There may be a few cells in D that show overlap of SOX2 and TFAP2A, but if the authors want to claim that a transition zone exists, they need to produce stronger evidence. Figure 7 is more convincing but see the next point.

Thank you for this insightful comment. We now address the nature of the transitioning boundary cell population extensively in our other recent study (Sekulovski et al., 2023).

(7) Figure 7 further confuses the issue. A zone at either end of the epiblast is clearly positive for Sox2 and the two amnion markers, clearer than in Figure 5, but why does the marker DLX5 overlap with SOX2 in the embryo (7d) but not the model (7C)? Arguments regarding intermediate cell populations would be greatly strengthened by scRNA-seq data on the model system.

In our original manuscript, our DLX5 staining was performed at 48-hr post-BMP, at which SOX2 expression is absent in all cells. Our new analysis at the 24-hr timepoint now shows that DLX5 is expressed in SOX2+ cells (this is now presented in Fig. 7C).

As stated in the point #6, our recent study comprehensively describes the transcriptomic and spatial characteristics of the transitioning boundary cell population (Sekulovski et al., 2023).

(8) L357 TFAP2C KO does not resemble intermediate cysts in Figure 2. In Figure 2, both SOX2 and amnion markers are co-expressed in the same cells. In 8C, SOX2 and ISL1 are mutually exclusive.

We agree with this comment, and now removed this statement pointing out the resemblance (Line#359 of the original manuscript).

(9) Figure 8d-the same caveats noted above regarding the interpretation of superposition of bulk RNA-seq data with scRNA-seq UMAP analysis apply here.

Please refer to our explanation in point#4.

**Reviewer #3:**
In this work, the authors tried to profile time-dependent changes in gene and protein expression during BMP-induced amnion differentiation from hPSCs. The authors depicted a GATA3 - TFAP2A - ISL1/HAND1 order of amniotic gene activation, which provides a more detailed temporary trajectory of amnion differentiation compared to previous works. As a primary goal of this study, the above temporal gene/protein activation order is amply supported by experimental data. However, the mechanistic insights on amniotic fate decision, as well as the transcriptomic analysis comparing amnion-like cells from this work and other works remain limited. While this work allows us to see more details of amnion differentiation and understand how different transcription factors were turned on in a sequence and might be useful for benchmarking the identity of amnion in ex utero cultured human embryos/embryoids, it provides limited insights on how amnion cells might diverge from primitive streak / mesoderm-like cells, despite some transcriptional similarity they shared, during early development.

We are happy that Reviewer #3 appreciates that our model can be used effectively to identify previously unrecognized amniotic gene activation cascade, providing a comprehensive timecourse transcriptomic resource.

As detailed below, we address specific concerns raised by Reviewer #3. We now provide additional mechanistic insights into amnion fate progression, and include additional transcriptomic comparisons with a cynomolgus macaque single cell RNA sequencing dataset.

**Reviewer #3 (Recommendations For The Authors):**
(1) The authors generated KO cell lines lacking GATA3 and TFAP2A, respectively. Their results showed some disrupted amnion differentiation only in TFAP2A-KO. Therefore, these data do not provide sufficient evidence to support whether these transcription factors are crucial for amnion fate specification. Perhaps an experiment could be done with overexpression of these markers and testing if they could force hPSC to adopt amnion-like fate.

Thank you for this insightful comment. We generated cell lines that enable us to inducibly express GATA3 or TFAP2A, and the transgene expression was induced at d2 (when BMP treatment is normally initiated) until d4. However, this inducible expression did not lead to amniogenesis, and cysts maintained pluripotency. Due to the uninterpretable nature, these results are not included in the revised manuscript.

As detailed extensively in the manuscript, within each cyst, amniogenesis is initially seen focally, then spreads laterally resulting in fully squamous amnion cysts. This is also seen in our previously published Gel-3D amnion model (extensively described in (Shao et al., 2017)). In the absence of TFAP2A, we showed that the focal amniogenesis is observed, but spreading is not seen, suggesting that TFAP2A controls amnion fate progression. Therefore, while TFAP2A is not critical for the amnion fate specification in the focal cells, our results show that TFAP2A indeed helps to promote amniotic specification of cells neighboring the focal amniotic cells. Moreover, in the revised manuscript, we now show that TFAP2A transgene expression in the TFAP2A-KO background restores formation of fully squamous hPSC-amnion, further establishing the role of TFAP2A in amnion fate progression (Fig. 8C of the revised manuscript, Line#362-364).

(2) The transcriptomic analysis made by the authors provides some comparison between BMPinduced amnion-like cells in vitro and the amnion-like cells from CS7 human embryo in vivo. However, the data set from the human embryo contains only a limited number of cells, and might not provide a sufficient base for decisive assessment of the true identity of amnion-like cells obtained in vitro. It might help if the authors could integrate their bulk sequencing data with other primate embryo data sets.

Thank you for this helpful comment. We have now performed our transcriptional similarity analysis using early (day 14) cynomolgus macaque embryo datasets generated in a study by (Yang et al., 2021), and found that the bulk time-course transcriptome of our hPSC-amnion model overlaps with the cynomolgus macaque amniotic lineage progression (Fig. 4F, Line#265268). We also now provide the expression of key markers within the Yang et al. dataset (GATA3, TFAP2A, ISL1, TBXT, DLX5, Fig. 4G, S2F).

(3) Following the point above, the authors used transcriptomic analysis to identify several intermediate states of cells during amnion differentiation and claimed that there is a primitivestreak-like intermediate. However, this might be an overstatement. During stem cell culture and differentiation, intermediate states showing a mixture of biomarkers are very common and do not imply that such intermediates have any biological meaning. However, stating that amnion differentiation passes through primitive streak-like intermediates, might imply a certain connection between these two lineages, for which there is a lack of solid support. Instead, a more interesting question might be how amnion and primitive streak differentiation, despite some transcriptomic similarity, diverge from each other during early development. What factors make this difference? The authors might further analyze RNA-seq data to provide some insights.

Thank you very much for the insightful comments.

We understand Reviewer #3’s concern that the intermediate state that we see may not recapitulate a primitive streak-like state. However, in our original manuscript, we described these cells as “Primitive Streak-like” because those cells were annotated as Primitive Streak in the dataset by Tyser et al. Interestingly, a recent study by Ton et al. showed that the Tyser et al. Primitive Streak cells actually consist of a mixture of different cell lineages (e.g., Epiblast, Nonneural ectoderm, Anterior or caudal primitive streak, PGC (Ton et al., 2023)). Therefore, we agree that it was an overstatement to call them “Primitive Streak-like”, and, to avoid confusions, we now label the TBXTlow sub-population found in the Tyser et al. Primitive Streak population as “TBXTlow state” throughout the manuscript.

Our data indicate that TFAP2A may play a role in controlling the lineage decision between amnion and primitive streak cells that abundantly express TBXT (TBXThigh). In the original manuscript, we included data showing that 48-hr TFAP2A-KO cysts show transcriptomic characteristics similar to some Primitive Streak cells (Fig. 8D). Intriguingly, our new data show that, in the absence of TFAP2A, some TBXThigh cells are indeed seen (Fig. 8E, Line#373-374). These results provide a body of evidence for the role of TFAP2A in promoting the amniotic lineage, perhaps by suppressing the TBXThigh state. This point is now addressed in the Discussion (Line#401-409).

Additional new data:

Using Western blot, we now show that GATA3 is absent in the GATA3-KO lines (Fig. S4C). We noticed that this was lacking in the original manuscript.

We now show that an inducible expression of TFAP2A in the TFAP2A-KO cysts leads to controllike cysts (Fig. 8C, Line#362-364).

Additional changes:

Typos were fixed in Fig. 5I – “boundary” and “disseminating” were not spelled correctly.

Line#350 – we originally noted “GATA3 expression precedes TFAP2A expression by approximately 12 hours”. This was incorrect, and is changed to 9 hours in the revised manuscript. We apologize for this mistake.

REFERENCES

Blakeley, P., Fogarty, N.M., del Valle, I., Wamaitha, S.E., Hu, T.X., Elder, K., Snell, P., Christie, L., Robson, P., and Niakan, K.K. (2015). Defining the three cell lineages of the human blastocyst by single-cell RNA-seq. Development 142, 3151-3165.

Castillo-Venzor, A., Penfold, C.A., Morgan, M.D., Tang, W.W., Kobayashi, T., Wong, F.C., Bergmann, S., Slatery, E., Boroviak, T.E., Marioni, J.C., et al. (2023). Origin and segregation of the human germline. Life Sci Alliance 6.

Granja, J.M., Klemm, S., McGinnis, L.M., Kathiria, A.S., Mezger, A., Corces, M.R., Parks, B., Gars, E., Liedtke, M., Zheng, G.X.Y., et al. (2019). Single-cell multiomic analysis identifies regulatory programs in mixed-phenotype acute leukemia. Nature biotechnology 37, 1458-1465. Meistermann, D., Bruneau, A., Loubersac, S., Reignier, A., Firmin, J., Francois-Campion, V., Kilens, S., Lelievre, Y., Lammers, J., Feyeux, M., et al. (2021). Integrated pseudotime analysis of human pre-implantation embryo single-cell transcriptomes reveals the dynamics of lineage specification. Cell stem cell 28, 1625-1640 e1626.

Ohgushi, M., Taniyama, N., Vandenbon, A., and Eiraku, M. (2022). Delamination of trophoblastlike syncytia from the amniotic ectodermal analogue in human primed embryonic stem cellbased differentiation model. Cell reports 39, 110973.

Okae, H., Toh, H., Sato, T., Hiura, H., Takahashi, S., Shirane, K., Kabayama, Y., Suyama, M., Sasaki, H., and Arima, T. (2018). Derivation of Human Trophoblast Stem Cells. Cell stem cell 22, 50-63 e56.

Petropoulos, S., Edsgard, D., Reinius, B., Deng, Q., Panula, S.P., Codeluppi, S., Plaza Reyes, A., Linnarsson, S., Sandberg, R., and Lanner, F. (2016). Single-Cell RNA-Seq Reveals Lineage and X Chromosome Dynamics in Human Preimplantation Embryos. Cell 165, 1012-1026.

Sasaki, K., Nakamura, T., Okamoto, I., Yabuta, Y., Iwatani, C., Tsuchiya, H., Seita, Y., Nakamura, S., Shiraki, N., Takakuwa, T., et al. (2016). The Germ Cell Fate of Cynomolgus Monkeys Is Specified in the Nascent Amnion. Developmental cell 39, 169-185.

Sekulovski, N., Juga, L.L., Cortez, C.L., Czerwinski, M., Whorton, A.E., Spence, J.R., Schmidt, J.K., Golos, T.G., Gumucio, D.L., Lin, C.-W., et al. (2023). Identification of amnion progenitor-like cells at the amnion-epiblast bounday in the primate peri-gastrula. bioRxiv doi:

10.1101/2023.09.07.556553.

Shao, Y., Taniguchi, K., Townshend, R.F., Miki, T., Gumucio, D.L., and Fu, J. (2017). A pluripotent stem cell-based model for post-implantation human amniotic sac development. Nature communications 8, 208.

Ton, M.N., Keitley, D., Theeuwes, B., Guibentif, C., Ahnfelt-Ronne, J., Andreassen, T.K., Calero-Nieto, F.J., Imaz-Rosshandler, I., Pijuan-Sala, B., Nichols, J., et al. (2023). An atlas of rabbit development as a model for single-cell comparative genomics. Nature cell biology 25, 10611072.

Tyser, R.C.V., Mahammadov, E., Nakanoh, S., Vallier, L., Scialdone, A., and Srinivas, S. (2021). Single-cell transcriptomic characterization of a gastrulating human embryo. Nature 600, 285289.

Yabe, S., Alexenko, A.P., Amita, M., Yang, Y., Schust, D.J., Sadovsky, Y., Ezashi, T., and Roberts, R.M. (2016). Comparison of syncytiotrophoblast generated from human embryonic stem cells and from term placentas. Proceedings of the National Academy of Sciences of the United States of America 113, E2598-2607.

Yang, R., Goedel, A., Kang, Y., Si, C., Chu, C., Zheng, Y., Chen, Z., Gruber, P.J., Xiao, Y., Zhou, C., et al. (2021). Amnion signals are essential for mesoderm formation in primates. Nature communications 12, 5126.